# Single molecule analysis reveals monomeric XPA bends DNA and undergoes episodic linear diffusion during damage search

Emily C. Beckwitt[1,2], Sunbok Jang[2,3], Isadora Carnaval Detweiler[4], Jochen Kuper[5], Florian Sauer [5], Nina Simon[6], Johanna Bretzler[6], Simon C. Watkins [7], Thomas Carell [6], Caroline Kisker[5] & Bennett Van Houten [2,3]✉

Nucleotide excision repair (NER) removes a wide range of DNA lesions, including UV-induced photoproducts and bulky base adducts. XPA is an essential protein in eukaryotic NER, although reports about its stoichiometry and role in damage recognition are controversial. Here, by PeakForce Tapping atomic force microscopy, we show that human XPA binds and bends DNA by ∼60° as a monomer. Furthermore, we observe XPA specificity for the helix-distorting base adduct N-(2'-deoxyguanosin-8-yl)-2-acetylaminofluorene over non-damaged dsDNA. Moreover, single molecule fluorescence microscopy reveals that DNA-bound XPA exhibits multiple modes of linear diffusion between paused phases. The presence of DNA damage increases the frequency of pausing. Truncated XPA, lacking the intrinsically disordered N- and C-termini, loses specificity for DNA lesions and shows less pausing on damaged DNA. Our data are consistent with a working model in which monomeric XPA bends DNA, displays episodic phases of linear diffusion along DNA, and pauses in response to DNA damage.

[1] Program in Molecular Biophysics and Structural Biology, University of Pittsburgh, Pittsburgh, PA 15260, USA. [2] UPMC Hillman Cancer Center, Pittsburgh, PA 15213, USA. [3] Department of Pharmacology and Chemical Biology, University of Pittsburgh, Pittsburgh, PA 15261, USA. [4] Department of Chemical Engineering, University of Pittsburgh, Pittsburgh, PA 15261, USA. [5] Rudolf Virchow Center for Experimental Biomedicine, Institute for Structural Biology, University of Würzburg, 97080 Würzburg, Germany. [6] Center for Integrated Protein Science at the Department of Chemistry, Ludwig Maximillian University of Munich, 81377 Munich, Germany. [7] Center for Biologic Imaging, University of Pittsburgh, Pittsburgh, PA 15261, USA. ✉email: vanhoutenb@upmc.edu

Nucleotide excision repair (NER) is a highly conserved DNA repair pathway that, through a series of damage recognition and verification steps, is able to act on a wide range of DNA lesions. In humans, this process involves approximately 30 proteins, working together to protect our genomes from the damaging effects of UV radiation and chemical agents like cisplatin, polycyclic aromatic hydrocarbons, and aromatic amines. UV radiation causes formation of two major lesions in DNA: the cyclobutane pyrimidine dimer (CPD) and 6-4 photoproduct (6-4PP), at a ratio of approximately three to one[1]. NER can be initiated in two general ways[2,3]. During transcription-coupled (TC) NER, RNA polymerase stalls at a lesion and is recognized by CSA and CSB, which promote removal of the polymerase from the damage site and recruitment of subsequent repair proteins. Recently, broader roles for CSA and CSB in proteasome-mediated degradation of an immediate early gene product and transcription restart after UV have been reported[4]. Global genome (GG) NER is initiated by UV-DDB and/or XPC-RAD23B when they recognize the presence of a lesion at any site in the chromatin. The two pathways converge for a damage verification step, where the TFIIH helicase complex unwinds the DNA and ensures that damage is present. The endonucleases XPF-ERCC1 and XPG cut the DNA sequentially, removing 24–32 bases while DNA polymerase (δ, ε, or κ) and ligase (I or III) fill in the gap.

XPA is a protein that plays an essential role in both TC and GG NER. Patients with reported mutations in the *XPA* gene experience some of the most severe phenotypes of a disease called xeroderma pigmentosum (XP) and are at a significantly increased risk for skin cancer[5]. The exact function of XPA in NER remains controversial, as evidence exists for multiple roles. Relative to non-damaged dsDNA, XPA displays specificity for UV-irradiated DNA[6], 6-4PPs[7], N-(2'-deoxyguanosin-8-yl)-2-acetylaminofluorene (dG-C8-AAF)[8–10], cisplatin[11], non-hybridized bases[12,13], and other artificially distorted substrates[13,14]. XPA also binds preferentially to partially single-stranded DNA and forked substrates[15,16]. Early literature on the function of XPA in NER suggests that it is involved in initial steps of damage recognition[17,18]. Additional data place XPA at the damage verification step, interacting with and enhancing damage specificity of TFIIH[19–21], or even later in the pathway[22], acting as a scaffold and interacting with other NER proteins[23], including RPA[24]. Importantly, it has been suggested that damage recognition in NER, which needs to accommodate a diverse range of structures, is accomplished via a "discrimination cascade" involving multiple proteins with imperfect selectivity[7,25,26]. In support of this model, XPA enhances the damage specificity of TFIIH by promoting both its translocation along non-damaged DNA and stalling on damaged DNA[20]. As such, it remains of significant interest to investigate how XPA interacts with DNA lesions.

There is limited structural data for full-length XPA, due to large regions of conserved intrinsic disorder (Supplementary Fig. 1a), particularly in the N- and C-termini[27–29]. The intrinsically disordered regions of XPA may function in protein-DNA and/or protein-protein interactions[29]. Human XPA contains 273 amino acids with a molecular weight of 31.4 kDa. The minimal DNA-binding domain (DBD) was first identified by Tanaka and colleagues[11], and later expanded to include residues 98–239 (ref. [30]), covering about half of the total protein length and a zinc-finger motif. Available XPA structures include early solution NMR studies of human XPA DBD[31,32], a recent crystal structure of the extended human DBD[33], co-crystal structures of yeast Rad14 (XPA homolog) DBD on damaged DNA[10,34], and a cryo-electron microscopy structure of the complex formed between XPA, TFIIH, and DNA[21]. At present, there is no structural information regarding the N- and C-termini of XPA or its

homologs in the RCSB Protein Data Bank. The Rad14 DBD structures show that the protein can bind DNA as a dimer flanking the damage site and produces a 70° bend in the DNA[10,34]. This result supports previous studies which concluded that XPA binds DNA as a homodimer[8,9,35]. XPA stoichiometry both on and off of DNA remains controversial as others have reported monomeric binding[15,21,36], or that stoichiometry is dependent on the DNA substrate[16].

We therefore set out to resolve four fundamental issues regarding how XPA interacts with sites of damage in DNA: (a) specificity for DNA lesions embedded in long DNA substrates, (b) stoichiometry of binding, (c) induced DNA bending at lesions and/or non-specific sites, and (d) how XPA examines DNA for damage. First, atomic force microscopy (AFM) is used to assess binding specificity of lesions embedded in long DNA substrates, stoichiometry, and DNA bending. To accomplish this, we validate a new PeakForce Tapping® AFM mode for calculating the molecular weight of small proteins bound to DNA. Second, we use single molecule fluorescence microscopy to follow how quantum dot-labeled full-length and truncated XPA (lacking the intrinsically disordered domains) interrogate DNA for damage in real time. Based on these data, we propose a model of XPA episodic motion in which different conformational states of the protein are associated with different modes of DNA target search and the presence of helix-distorting DNA damage stabilizes tighter binding.

## Results

**XPA binds specifically to a dG-C8-AAF lesion.** We first confirmed that XPA recognizes AAF-adducted DNA by electrophoretic mobility shift assay (EMSA) as reported by others[8–10,12]. We generated binding isotherms of XPA by incubating a 37 bp DNA duplex (8 nM) with or without a single dG-C8-AAF adduct (AAF$_{37}$ and ND$_{37}$, respectively) and increasing amounts of purified full-length human XPA (His-flXPA-StrepII, Supplementary Fig. 1b). The apparent equilibrium dissociation constant ($K_D$) was $253 \pm 14$ nM (best fit value ± standard error of the fit) for ND$_{37}$ and $109 \pm 5$ nM for AAF$_{37}$, an approximately 2.3-fold difference (Supplementary Fig. 1c, e, g). Because AAF$_{37}$ contains only one specific site among 36 non-damaged bp, this difference can be multiplied by a factor of 37 to account for non-specific binding to the AAF$_{37}$ substrate[17,37]. This results in approximately 85-fold specificity for dG-C8-AAF over non-damaged DNA, in good agreement with a previously reported specificity of XPA for 6-4PP[17]. We also found the specificity for a CPD lesion to be ~44-fold (Supplementary Fig. 1d, i). Furthermore, at higher XPA concentrations, a second band of higher molecular weight appeared, indicating binding of an additional XPA protein. It is important to note that any affinity of XPA for DNA ends could obscure EMSA results in terms of (a) specificity, as end-binding (a common feature of DNA-binding proteins[38,39]) would increase overall binding on both substrates, thereby lowering the apparent specificity for the lesion, and (b) stoichiometry, as separate XPA proteins bound to the lesion and the end of the DNA would migrate the same as a true dimer in the gel. If XPA has fairly high non-specific binding affinity, then embedding a lesion in a long stretch of non-damaged DNA might pose difficulties in damage recognition.

We therefore turned to AFM to study XPA binding to a 538 bp DNA substrate with or without a single dG-C8-AAF lesion. The small size of XPA presents challenges in terms of resolution and ability to visualize XPA bound to DNA using AFM in tapping mode; we thus adopted the use of PeakForce Tapping mode (Bruker) with a 2 nm tip to achieve improved resolution and reduced sample deformation[40,41]. Using this method, we were

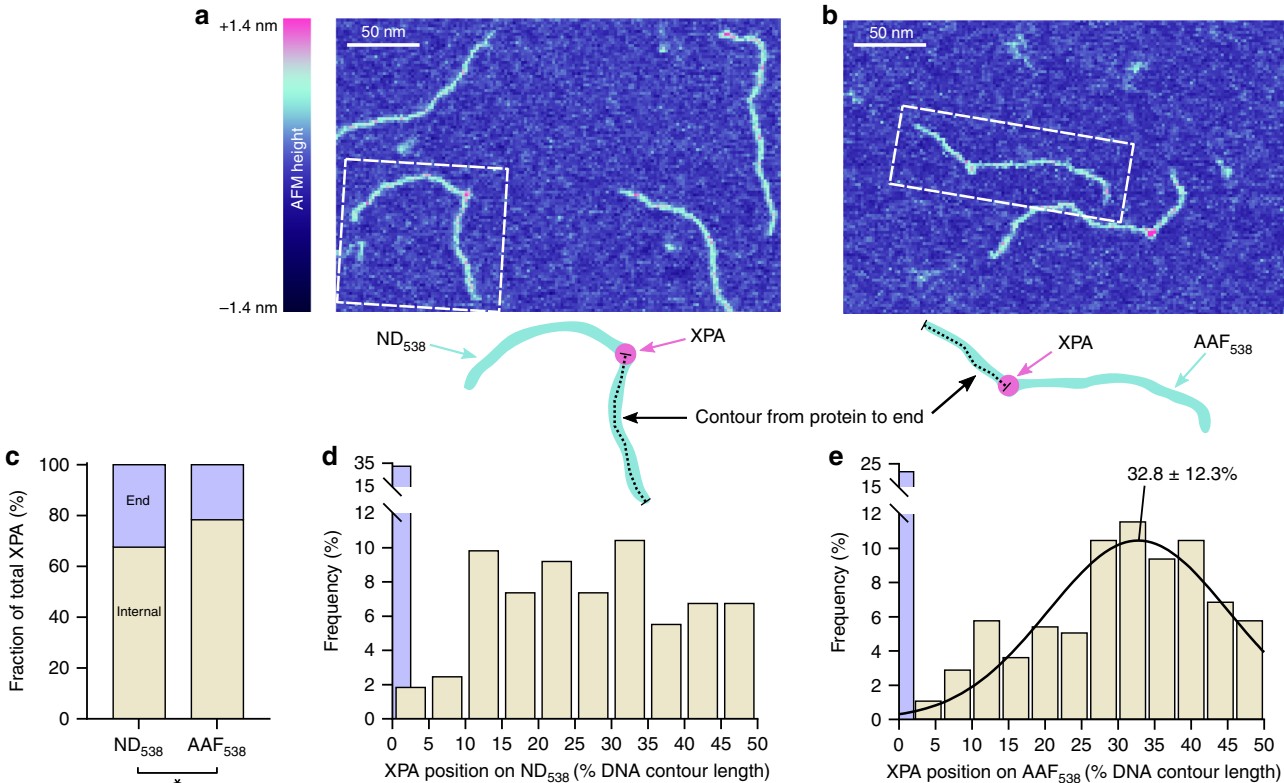

**Fig. 1 XPA binds specifically to a dG-C8-AAF lesion. a, b** Representative AFM image of XPA bound to a non-damaged 538 bp DNA substrate (**a**, ND$_{538}$) or a 538 bp DNA with a single dG-C8-AAF lesion at 30% from one end (**b**, AAF$_{538}$). The dashed white line indicates the example in the cartoon below. Binding position was measured between the center of the protein to the closest DNA end as a percentage of total DNA contour length. **c** Percentage of XPA bound to DNA at ends (lavender) or internally (tan) on ND$_{538}$ ($n = 163$ particles) and AAF$_{538}$ ($n = 277$ particles). *$p = 0.0118$ by $\chi^2$ test. **d** Histogram showing the distribution of internally bound XPA ($n = 110$ particles) position on ND$_{538}$. End-binders are shown in lavender. **e** Histogram and Gaussian fitting of internally bound XPA ($n = 217$ particles) positions on AAF$_{538}$. End binders are shown in lavender (not included in Gaussian fit). Gaussian is labeled with mean and s.d.

able to clearly recognize XPA bound to DNA by the increased AFM height and width of the complex (Fig. 1a, b).

Full-length human XPA (His-flXPA, Supplementary Fig. 1b, 1–4 µM) was incubated with a non-damaged 538 bp DNA substrate (ND$_{538}$, 100 nM) and three-dimensional images were obtained using PeakForce Tapping AFM. Of all the complexes observed, 33% were bound to the ends of the DNA substrate (Fig. 1c); we observed no instances of XPA bound to both ends. For the remaining internally-bound proteins, position along the DNA molecule was measured as a percentage of the total contour length of the DNA, which revealed no preference for a specific internal site (Fig. 1d).

XPA (His-flXPA) was then incubated with a 538 bp DNA substrate of the same sequence as ND$_{538}$ but with a single dG-C8-AAF lesion at 30% from the 3′ end (AAF$_{538}$). On this substrate, only 22% of bound XPA proteins were found at the DNA ends. We also observed an increased frequency of complexes found near the lesion, at the expense of end-binders and other non-specific complexes (Fig. 1e). A Gaussian was fit to the distribution of binding positions with a mean of 32.8 ± 12.3% (mean ± s.d. of Gaussian). This spread of values is similar to others we have reported for lesion-binding proteins[42]. Using the method of Erie and colleagues to assess specificity without confounding end-binders[38], we find that XPA has a specificity for dG-C8-AAF of ~660 (Supplementary Note 1, Supplementary Fig. 2). Based on these data, it is clear that XPA is able to bind non-specifically to DNA (at ends and non-damaged sequences). However, the protein does exhibit specificity for an AAF adduct embedded in a long DNA substrate and binds preferentially at such a site.

**XPA is a monomer in the absence of DNA**. To investigate the question of XPA stoichiometry, we first sought to clarify the oligomeric status of the free protein. While there is support that XPA is a monomer in solution[43–45], there have also been reports that it forms dimers and higher oligomers[35]. AFM has been successfully used to determine protein stoichiometry due to the linear relationship between AFM volumes of globular proteins and their molecular weight[46,47]. Full-length human XPA (His-flXPA, 32.6 kDa) was diluted to 40 nM and deposited on mica for imaging by PeakForce Tapping AFM in air. Volumes of the particles were measured and the data fit a Gaussian distribution centered at 30.3 ± 15.4 nm$^3$ (Fig. 2a).

In order to translate this volume into molecular weight, and thus protein stoichiometry, we generated a standard curve using monomeric proteins of known sizes (Fig. 2b). HMGB1 (25 kDa), APE1 (37 kDa with His tag), DNA polymerase β (Polβ, 42.8 kDa with His tag), and UvrD (85.6 kDa with His tag) were adsorbed at 40 nM on mica and imaged by PeakForce Tapping AFM (Supplementary Fig. 3). Measured AFM volumes of these proteins were plotted against their known molecular weight and fit using least-squares linear regression: Volume = (MW × 1.14) − 2.00. Using this equation, the measured volumes for XPA correspond to 28.4 ± 15.3 kDa, close to the expected molecular weight of 32.6 kDa for the monomer. There was no significant population of XPA corresponding to the dimer size.

To confirm that XPA is a monomer at higher protein concentrations, we performed size exclusion chromatography coupled with multiangle light scattering (SEC-MALS). At 65 µM and 80 µM, purified XPA (His-flXPA-StrepII, 33.9 kDa) eluted in

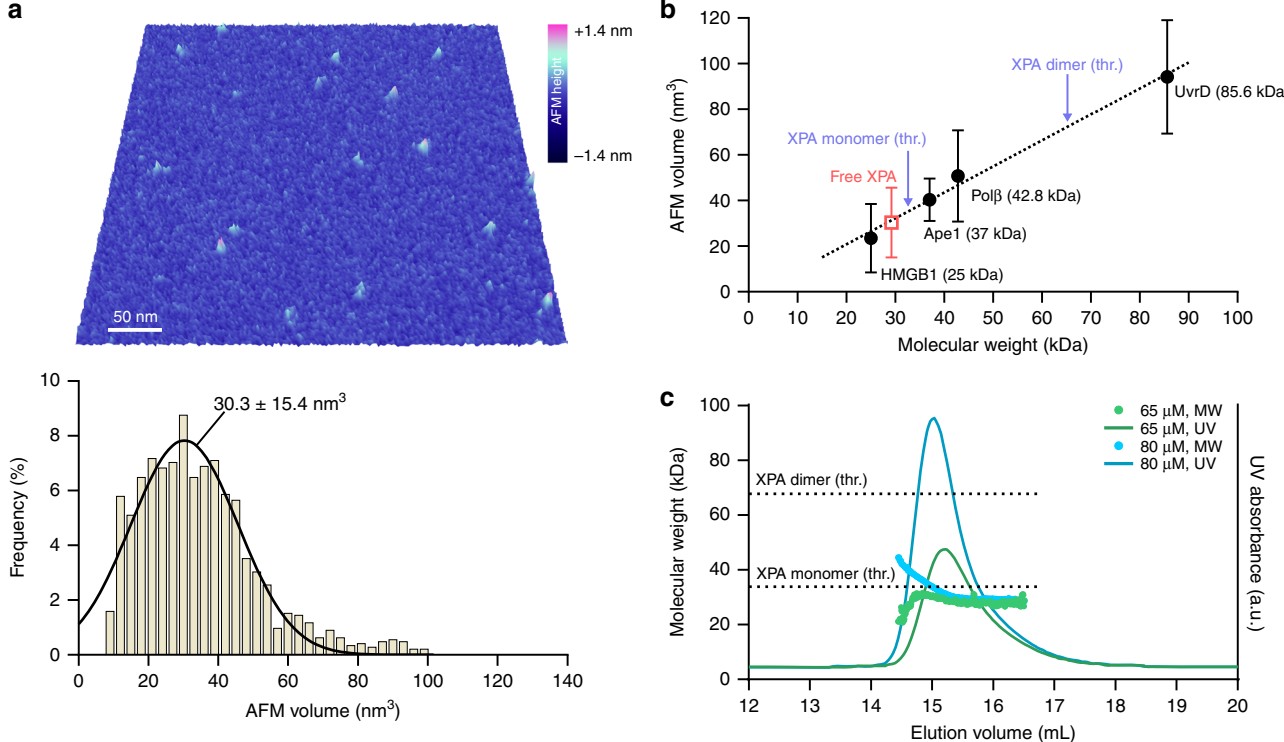

**Fig. 2 XPA is a monomer in the absence of DNA. a** Top, representative 3D AFM image of free XPA. Bottom, histogram and Gaussian fit of free XPA molecules imaged by AFM ($n = 1451$ particles). Gaussian is labeled with mean and s.d. **b** Standard for calculating molecular weight from AFM volumes. Volumes of free proteins imaged by PeakForce Tapping AFM were calculated and respective histograms were fit by Gaussian distributions. Solid black circles, proteins used to generate the standard plotted against their known MW (results and $n$ values for each in Supplementary Fig. 3). Dashed line, linear regression, resulting in the calibration curve: Volume ($nm^3$) = $1.14 \times$ MW (kDa) – 2.00. $R^2 = 0.990$. Purple arrows point to theoretical (thr.) volumes for the purified His-flXPA monomer (32.6 kDa) and dimer (65.2 kDa). Open red square, experimental His-flXPA AFM volume (see a), corresponding to a molecular weight of $28.4 \pm 15.3$ kDa. Errors bars indicate s.d. of the Gaussian distribution. **c** Molecular weight (MW) determination by SEC-MALS of XPA at 65 μM (green) and 80 μM (blue). Theoretical molecular weights for the purified His-flXPA-StrepII monomer (33.9 kDa) and dimer (67.8 kDa) are indicated by dashed lines.

a single major peak, with a molecular weight corresponding principally to a monomer (Fig. 2c). By both methods, the apparent molecular weight was a few kDa less than the theoretical value based on the sequence with tags, potentially due to protein conformation. Together, these data clearly indicate that XPA (40 nM–80 μM) predominantly exists as a monomeric species in solution.

**XPA binds non-damaged DNA and dG-C8-AAF as a monomer.** Previous reports suggest that XPA may bind DNA as a homodimer[9,35] while others indicate monomeric binding[15,21,36]. AFM offers the unique ability to distinguish between distinct complexes (e.g., one protein at a DNA end and one protein at a lesion) and true dimerization. Therefore, we measured the AFM volumes of XPA bound to DNA in order to determine stoichiometry. Because XPA is a small protein, we expected the DNA to contribute a significant amount of volume to the total complex. Based on published studies[48], we developed a method to determine the size of the DNA within the complex and subtract its volume to obtain the volume of XPA alone, and validated this using two proteins known to bind DNA as monomers, APE1 and Polβ (Supplementary Note 2, Supplementary Fig. 4).

Having confirmed that we would be able to distinguish XPA monomers and dimers on DNA, we measured AFM volumes for XPA (His-flXPA, 32.6 kDa) bound to ND_538 and AAF_538 (Fig. 3). In some cases, we observed multiple binding events on the same DNA molecule; these were measured individually if there was a clear stretch of unbound DNA between them. XPA bound to

ND_538 had a distribution of AFM volumes centered at $28.4 \pm 12.7$ $nm^3$ (mean ± s.d. of the Gaussian), corresponding to $26.7 \pm 12.9$ kDa (Fig. 3b). The volumes of internally and end-binding proteins had similar distributions. XPA bound to AAF_538 had a distribution of volumes centered at $31.0 \pm 12.9$ $nm^3$, corresponding to $29.0 \pm 13.1$ kDa (Fig. 3d). Again, the volumes of internally-binding (either near the lesion or not) and end-binding proteins had similar distributions. Thus, our AFM data show that, regardless of where XPA binds DNA, it does so as a monomer. We did not observe a significant population corresponding to a dimer.

**XPA bends DNA ~60°.** Having shown that XPA binds to damaged and non-damaged DNA as a monomer, we next asked whether XPA induces a bend at the dG-C8-AAF site. We first sought to determine if the dG-C8-AAF lesion itself introduces any flexibility into the DNA helix. Because this lesion was placed at 30% from the 3' end of the AAF_538 substrate, we measured the DNA bend angle at this site for unbound DNA. Note that while there is a single lesion in our damaged substrate, we measured the angle at 30% from both ends (resulting in two angles per DNA molecule) because, under the current conditions, we are unable to differentiate between the 5' and 3' end of the molecule. As a negative control we measured the DNA angle at 30% from each end of ND_538 (Fig. 4a), and found a distribution centered around 0° (Fig. 4c). The distribution for the dG-C8-AAF-modified DNA, AAF_538, showed the emergence of a second population of bend angles (Fig. 4b, d). A double Gaussian fit to the data describes two

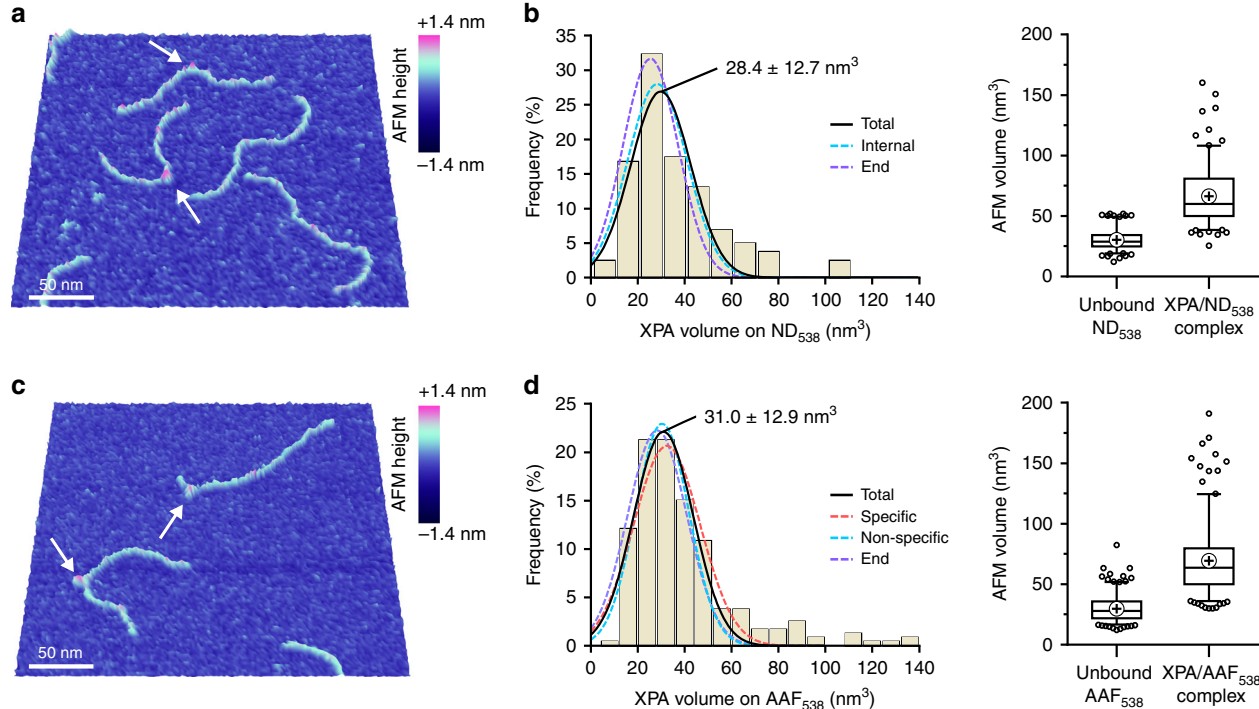

**Fig. 3 XPA binds non-damaged DNA and dG-C8-AAF modified DNA as a monomer. a, c** Representative 3D AFM image of XPA bound to 538 bp non-damaged DNA (**a**, $ND_{538}$) or 538 bp DNA with a single dG-C8-AAF adduct at 30% from one end (**c**, $AAF_{538}$). White arrows point to XPA bound to DNA. **b** Left, histogram showing distribution of AFM volumes of all XPA molecules bound to $ND_{538}$ ($n = 161$ particles) with Gaussian fit (solid black line, labeled with mean ± s.d. of the distribution). AFM volume corresponds to 26.7 ± 12.9 kDa, using the standard shown in Fig. 2b. Dashed cyan line, Gaussian fit to the sub-fraction of non-specifically bound XPA (i.e., all internal complexes, $n = 108$); centered at 29.9 nm$^3$ (s.d. 12.8 nm$^3$), corresponding to 28.0 ± 13.0 kDa. Dashed purple line, Gaussian fit to the sub-fraction of XPA bound at DNA ends ($n = 53$); centered at 25.5 nm$^3$ (s.d. 11.7 nm$^3$), corresponding to 24.1 ± 12.0 kDa. Right, box and whisker plots (5-95 percentile) of unbound DNA and total complex volume measurements. **d** Left, histogram showing the distribution of AFM volumes of all XPA proteins ($n = 235$ particles) bound to $AAF_{538}$ with a Gaussian fit (solid black line). The AFM volume corresponds to 29.0 ± 13.1 kDa. Dashed red line, Gaussian fit to the sub-fraction of XPA bound between 20 and 40% of the DNA contour length ("specific," $n = 58$); centered at 32.4 nm$^3$ (s.d. 14.4 nm$^3$), corresponding to 30.2 ± 14.4 kDa. Dashed cyan line, Gaussian fit to the sub-fraction of XPA bound internally but at positions away from the lesion ("non-specific," $n = 84$); centered at 30.5 nm$^3$ (s.d. 11.4 nm$^3$), corresponding to 28.5 ± 11.8 kDa. Dashed purple line, Gaussian fit to the sub-fraction of XPA bound at DNA ends ($n = 51$); centered at 28.3 nm$^3$ (s.d. 13.0 nm$^3$), corresponding to 26.6 ± 13.2 kDa. Right, box and whisker plots (5-95 percentile) of unbound DNA and total complex volume measurements.

populations at 10.5 ± 7.0° and 34.8 ± 10.6°. These data are consistent with the smaller angle being the unmodified site and the larger angle being the result of flexibility introduced by dG-C8-AAF[49,50].

We next asked if XPA bends the DNA. Our AFM data show that XPA bends both non-damaged (Fig. 4e) and AAF-adducted (Fig. 4f) DNA. DNA angles were measured at all sites of internally-bound XPA. On $ND_{538}$, XPA induced a bend angle of 54.0 ± 30.1° (Fig. 4g). On $AAF_{538}$, XPA induced a bend angle of 58.6 ± 26.8°; the distribution was essentially the same for XPA bound near the lesion or non-specifically (Fig. 4h). The angle is greater than that introduced by the lesion itself (~30°). Together, our AFM data indicate that XPA binds preferentially to the helix-bending dG-C8-AAF site and that, regardless of binding site, the complex contains a single XPA protein and the DNA is bent ~60°.

**XPA performs episodic 1D diffusion to search DNA for damage.** The width of the Gaussian distribution of XPA binding position on $AAF_{538}$ suggests that the protein might be dynamic on DNA and migrate some distance away from a damaged site. We have previously shown that Rad4-Rad23, which adopts a similar distribution of binding positions by AFM, performs constrained linear diffusion around a lesion[42]. Therefore, we next investigated whether XPA displays one-dimensional diffusion on

non-damaged and damaged DNA using a single molecule DNA tightrope assay[51,52].

DNA tightropes consisted of long (>40 kbp) DNA molecules suspended (elongated ~90% of the DNA contour length) between beads in a flow cell. His-tagged XPA was labeled with either a streptavidin-conjugated 705 nm quantum dot (Qdot) and biotinylated anti-His antibody (Fig. 5a) or an anti-mouse IgG antibody-conjugated 605 nm Qdot and mouse anti-His antibody. Both strategies resulted in similar observed behavior (Supplementary Fig. 5e). For observation, flow was turned off and 300 s movies were recorded at 10–12.5 frames per second (fps) of XPA on one of four different tightrope substrates: non-damaged genomic λ DNA (NDλ), UV-treated λ DNA (UVλ$_{20J}$ and UVλ$_{80J}$), and defined arrays of dG-C8-AAF (AAF$_{array}$). Exposure of λ DNA to 20 J/m$^2$ (UVλ$_{20J}$) or 80 J/m$^2$ (UVλ$_{80J}$) 254 nm UV-C radiation results in a lesion density of approximately one UV photoproduct every 2.2 kbp or 550 bp, respectively (ref. [53]). Because UV-C radiation leads to formation of lesions comprising ~75% CPDs (relatively non-distorting to DNA helix) and ~25% 6-4PPs (which induce a 44° bend in DNA)[1,54], we expect UVλ$_{80J}$ to contain one 6-4PP every ~2.2 kbp. AAF arrays were prepared via end-to-end ligation of a 2030 bp fragment of linear DNA with a single site-specific dG-C8-AAF modification[53,55]. Kymographs were first categorized into four groups: stationary/persistent, stationary/dissociated, motile/persistent, and motile/dissociated.

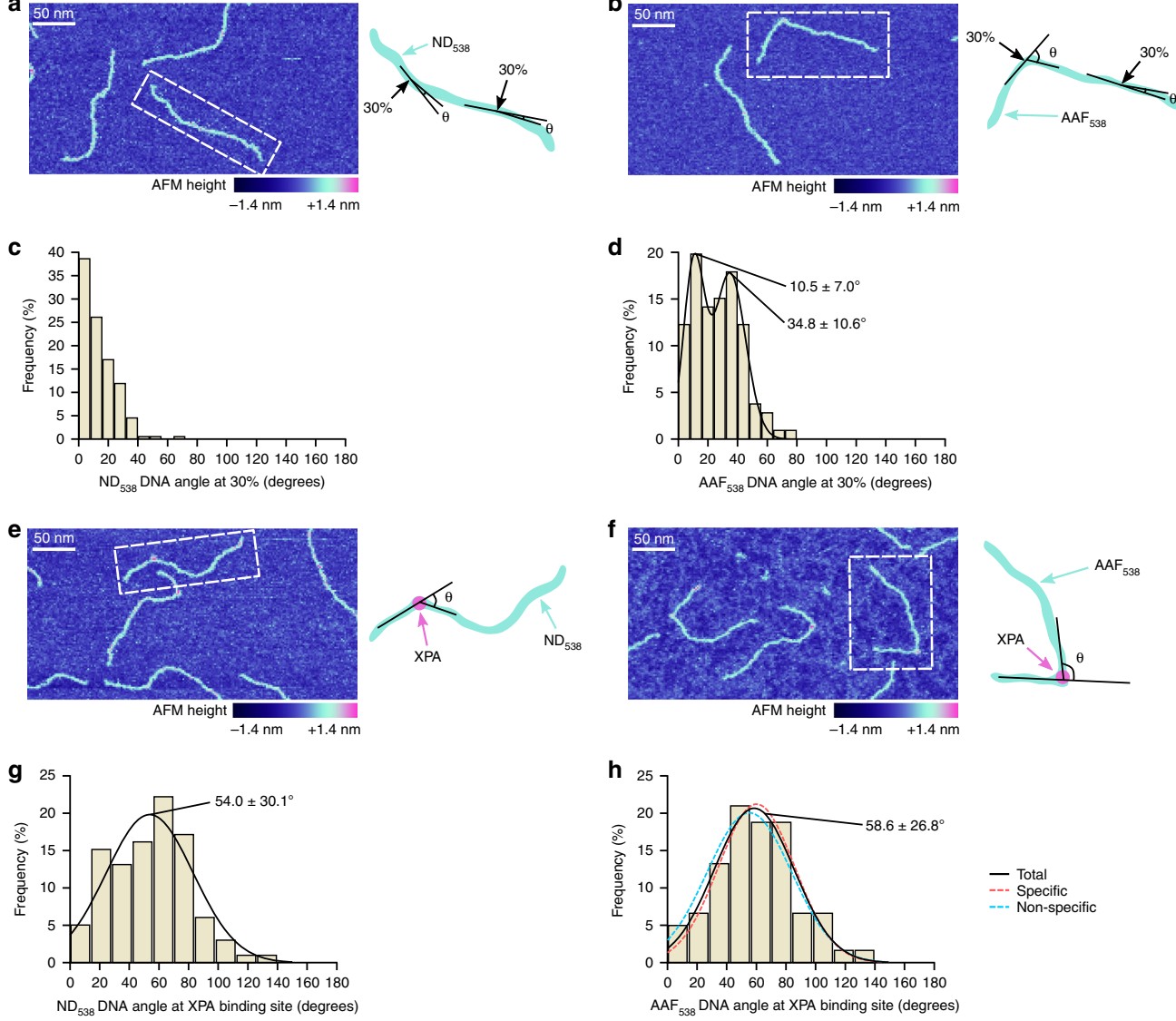

**Fig. 4 dG-C8-AAF bends DNA ~35° and XPA bends DNA ~60°. a, b** Representative AFM image of free 538 bp non-damaged DNA (**a**, ND$_{538}$) or free 538 bp DNA with a single dG-C8-AAF adduct at 30% from one end (**b**, AAF$_{538}$). Dashed white line indicates the example in the cartoon to the right. Angles were measured at sites 30% of the total DNA contour length from each end and are reported as the θ angle (supplement to internal DNA angle). **c** Histogram of the inherent DNA bend angle of non-damaged DNA ($n = 176$ angles). **d** Histogram and double Gaussian fitting of the inherent DNA bend angle of AAF-adducted DNA ($n = 106$ angles). Each peak of the Gaussian is labeled as mean ± s.d. **e, f** Representative AFM image of XPA bound to ND$_{538}$ (**e**) or AAF$_{538}$ (**f**). The dashed white line indicates the example in the cartoon to the right. Angles were measured at sites of bound XPA and are reported as the θ angle. **g** Histogram and Gaussian fitting showing the distribution of DNA bend angles at all sites of internally bound XPA on ND$_{538}$ ($n = 99$ angles). **h** Histogram showing distribution of DNA bend angles at all sites of internally bound XPA on AAF$_{538}$ ($n = 181$ angles) with Gaussian fit (solid black line). Gaussian is labeled as mean ± s.d. Dashed red line, Gaussian fit to the sub-fraction of angles at XPA sites between 20 and 40% of the DNA contour length ("specific," $n = 116$); centered at 60.2° (s.d. 25.7°). Dashed cyan line, Gaussian fit to the sub-fraction of angles at sites of XPA bound internally but at positions away from the lesion ("non-specific," $n = 65$); centered at 55.3° (s.d. 28.5°).

Motility was defined as linear displacement greater than 130 nm (~380 bp or three pixels, see Methods) over the course of observation. XPA was primarily stationary (60–70% of the molecules) on all tightrope substrates (Fig. 5b, flXPA). Although the proportions of these broad categories do not appear to be affected by the DNA substrate, we expected the nature of diffusion to differ depending on the type and lesion density. We therefore analyzed the motile fraction in greater detail.

Unlike other repair proteins we have observed at the single molecule level[42,53,56], motile XPA particles often changed their diffusive behavior multiple times during observation (Fig. 5c,

Supplementary Movie 1). Most exhibited some periods of pausing between episodic phases of linear diffusion. We categorized the behavior of motile XPA into three distinct modes: paused (particle not moving), short-range diffusion (displacement less than 690 nm), and long-range diffusion (displacement greater than 690 nm). Thresholds for motile modes were chosen based on approximate distances between lesions; assuming true B-form DNA, 2030 bp correspond to 690 nm. Fig. 5c shows an example of XPA on NDλ, exhibiting all three modes; the behavior of each particle was divided into phases based on these modes.

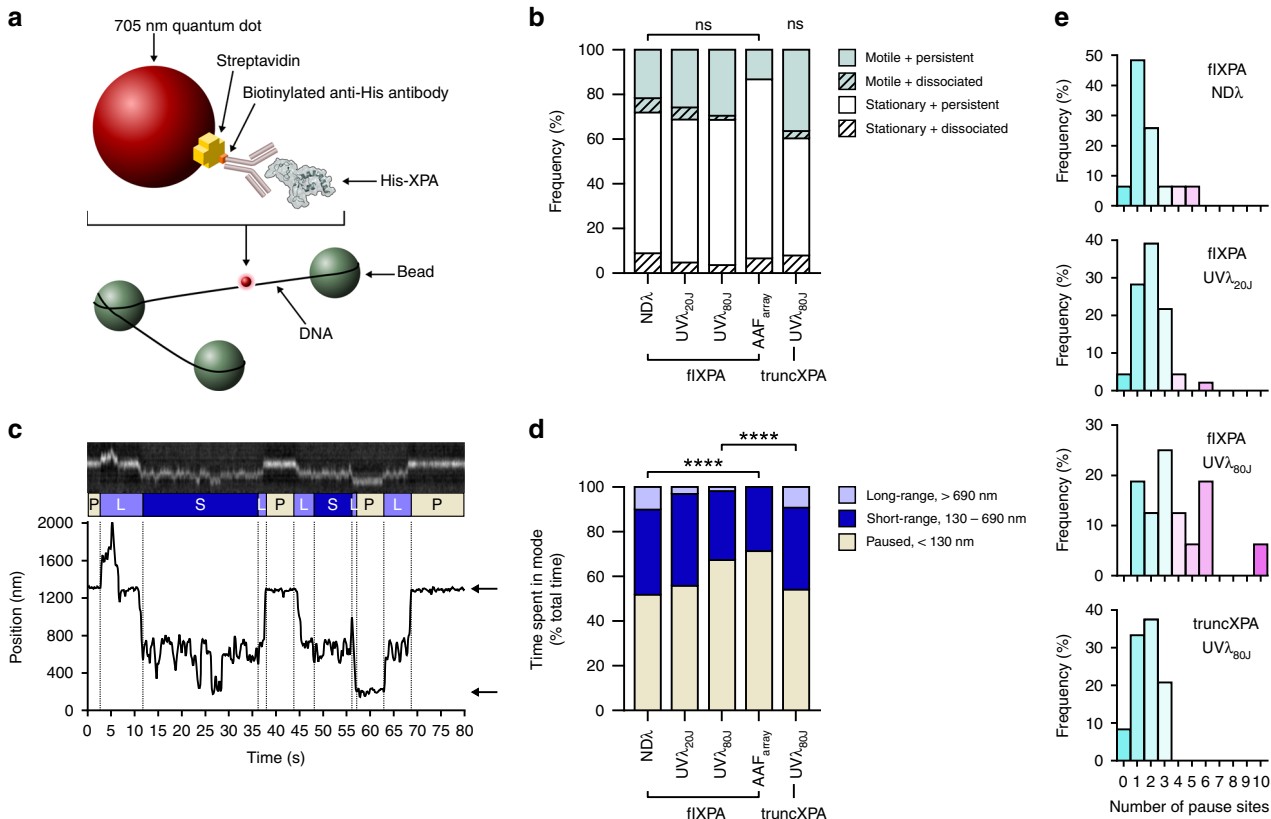

**Fig. 5 XPA exhibits episodic linear diffusion on DNA tightropes. DNA damage leads to increased pausing, dependent on N- and C-termini. a** Cartoon showing one strategy used for XPA labeling on DNA tightropes. His-tagged XPA is labeled with a biotinylated anti-His antibody bound to a streptavidin-conjugated 705 nm quantum dot. See Methods for alternative labeling strategy. **b** Stacked bar graph showing the fraction of motile (teal) vs. stationary (white) and persistent (solid) vs. dissociating (diagonal lines) particles of full-length XPA (flXPA) on non-damaged λ (NDλ, $n = 124$ particles), 20 J/m² UV-irradiated λ (UVλ$_{20J}$, $n = 147$), 80 J/m² UV-irradiated λ (UVλ$_{80J}$, $n = 54$), and AAF arrays (AAF$_{array}$, $n = 45$), and of truncated XPA (truncXPA) on UVλ$_{80J}$ ($n = 63$). Motile particles are defined as those which moved more than 130 nm on DNA during 300 s observation. ns, no significant difference between groups by $\chi^2$ for all flXPA experiments, for all flXPA and truncXPA categories, or for truncXPA on UVλ$_{80J}$ vs. flXPA on UVλ$_{80J}$. **c** Example of a kymograph (cut to show only 80 s of the recorded movie) of motile 705 nm quantum dot-labeled flXPA on NDλ. Particle position (bottom) was localized using Gaussian fittings to the intensity profile on the fluorescence image (top). Dashed lines separate phases and diffusive modes are labeled according to particle displacement: paused (P, displacement <130 nm, tan), short-range motion (S, displacement 130–690 nm, navy), and long-range motion (L, displacement > 690 nm, lavender). Arrows point to pause sites occupied by particle. **d** Stacked bar graph showing the fraction of time spent in each mode as percentage of total time recorded for all motile particles. ****$p < 0.0001$ by $\chi^2$ for all flXPA experiments and for truncXPA on UVλ$_{80J}$ vs. flXPA on UVλ$_{80J}$. **e** Histogram of number of pause sites for flXPA on NDλ ($n = 31$ particles), UVλ$_{20J}$ ($n = 46$), and UVλ$_{80J}$ ($n = 16$), and for truncXPA on UVλ$_{80J}$ ($n = 24$).

**DNA damage increases pausing in motile XPA particles**. The proportion of seconds spent in each mode was calculated as a fraction of total recorded time for motile particles (Fig. 5d). Full-length XPA (flXPA) demonstrated a clear dose-dependent increase in paused time with increasing damage. Motile particles of flXPA spent 52% of the time paused on NDλ tightropes; this increased to 56% on UVλ$_{20J}$, 67% on UVλ$_{80J}$, and 71% on AAF$_{array}$, and was accompanied by a decrease in time spent in the diffusive modes, especially long-range mode.

Motile XPA appeared to prefer and return to certain positions on the tightropes (Fig. 5c). These "pause sites" were defined as sites at which XPA spent at least one paused phase (≥5 s). On NDλ, the majority of XPA particles (48.4%) paused at just one position (Fig. 5e, top row). On UVλ$_{20J}$, the majority of XPA particles (39.1%) paused at two distinct positions (Fig. 5e, second row), and on UVλ$_{80J}$, the majority (25.0%) occupied 3 distinct positions, with some occupying as many as 10 positions (Fig. 5e, third row). The increasing number of pause sites observed with increasing UV lesion density in the tightropes strongly suggests that these positions correspond to UV lesions. Pause sites observed on non-damaged DNA might represent other distorted

regions of the DNA (e.g., A-tract-GC junctions, spontaneous damage, or transient melting of base pairs). Due to the small sample size of motile XPA particles on AAF$_{array}$ ($n = 6$), this dataset was not included in this and subsequent analyses.

**Truncated XPA exhibits reduced pausing on damaged DNA**. We hypothesized that the intrinsically disordered N-terminal and C-terminal arms of XPA may play an important role during target search. We therefore examined the behavior of a truncated XPA variant (truncXPA) comprising residues M98 through T239 (ref. [30]). Truncated XPA maintains its ability to bind non-damaged 37 bp DNA ($K_D = 269 \pm 22$ nM) with similar affinity as flXPA (Supplementary Fig. 1f). Specific binding of truncXPA to an AAF adduct ($K_D = 189 \pm 9$ nM) is ~53-fold (Supplementary Fig. 1h). Thus, we observed a reduction in specificity to dG-C8-AAF by a factor of 1.6 in the truncated XPA compared to full-length protein. All specificity was lost for CPD ($K_D = 269 \pm 24$ nM, Supplementary Fig. 1j).

On UVλ$_{80J}$ tightropes, truncXPA exhibited the same proportion of stationary/motile and persistent/dissociated particles as flXPA (Fig. 5b). Importantly, for motile particles, we observed a

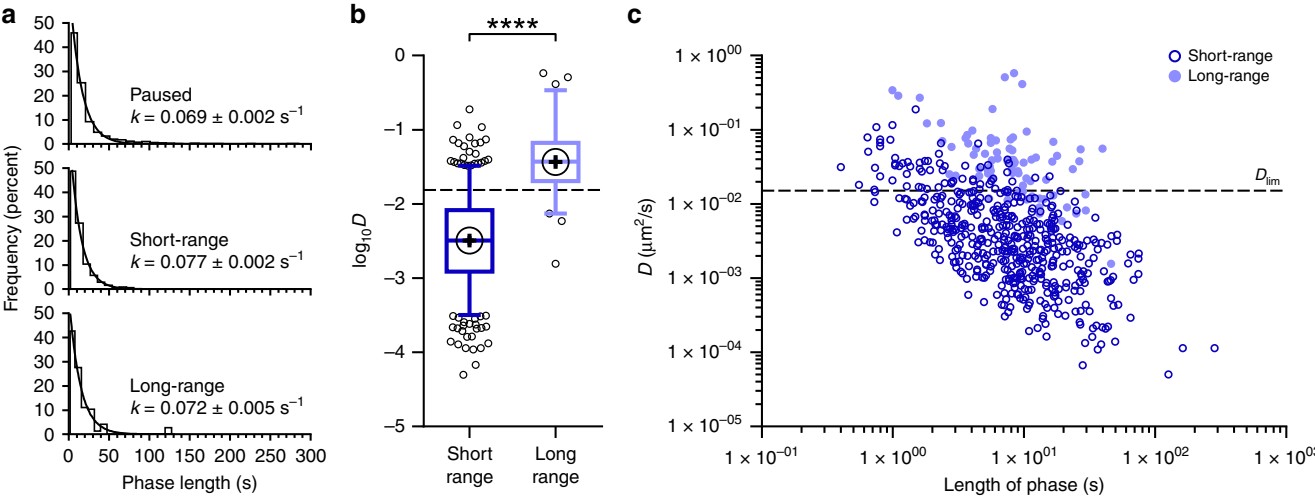

**Fig. 6 Short-range motion is associated with a lower diffusion coefficient. a** Cumulative frequency distributions of the lengths of paused ($n = 774$ phases), short-range ($n = 819$ phases), and long-range ($n = 145$ phases) phases for XPA on combined λ substrates (NDλ, UVλ$_{20J}$, and UVλ$_{80J}$). A single exponential is fit to each histogram and resulting $k$ is reported as best fit value ± s.e. of the fit. Mean lifetime, $\tau = 1/k$. **b** Box and whisker plot (5-95 percentile) of the diffusion coefficient ($\log_{10}D$) calculated for short-range ($n = 505$ phases) and long-range ($n = 79$ phases) phases of flXPA on combined λ substrates (NDλ, UVλ$_{20J}$, and UVλ$_{80J}$). +, sample mean. Dashed line, $D_{lim}$, theoretical limit to free diffusion for Qdot-flXPA. ****$p < 0.0001$ by two-tailed Student's $t$ test. **c** Plot of diffusion coefficient ($D$) vs. length of phase (same phases shown in **b**). Dashed line, $D_{lim}$, theoretical limit to $D$ for free diffusion of Qdot-flXPA.

significant decrease (~25%) in time spent paused and a corresponding increase in time spent undergoing long-range motion for truncXPA compared to flXPA on UVλ$_{80J}$ (Fig. 5d). Furthermore, the number of pause sites occupied by truncXPA on UVλ$_{80J}$ decreased dramatically (Fig. 5e). Hence, truncXPA behavior on highly damaged UV-irradiated DNA more closely resembled flXPA behavior on non-damaged DNA. These data suggest that while the core DNA-binding domain of XPA is capable of binding to damage and forming stationary complexes, its ability to transition into the paused state and form high affinity stable complexes is impaired without the N- and C-termini.

**XPA changes search mode on the seconds time scale.** We then considered the length of all individual phases to obtain an estimate for the phase lifetimes for each mode of motile XPA (full-length and truncated). A single exponential was fit to histograms of the phase lengths (Fig. 6a). The resulting mean lifetimes ($\tau$) are as follows: 14.4 s for paused phases, 13.1 s for short-range phases, and 14.0 s for long-range phases. Thus, within the defined limits of experimental temporal resolution, XPA changes its search state on the seconds time scale. Multiple comparisons of phase lengths between DNA substrates and/or protein size did not reveal any meaningful differences (Supplementary Fig. 6a). These data suggest that the time XPA molecules spend in any particular phase is not influenced by the presence of DNA damage or the disordered arms of the protein. It is also important to note that a significant proportion of XPA particles that were stationary during the entire 300 s observation window (Fig. 5b), may represent a distinct population of long-lived paused proteins.

**Long-range motion is associated with faster diffusion rates.** In order to gain insight into the rates of diffusion of motile XPA particles, we used mean squared displacement (MSD) analysis[55]. Because motile XPA changed its behavior so distinctly, generating MSD plots for the entire length of each kymograph was not appropriate. Instead, each diffusive phase was analyzed independently. The diffusion coefficient ($D$) is an order of magnitude

lower in the short-range mode ($3.2 \times 10^{-3}$ μm²/s) compared to long-range ($3.7 \times 10^{-2}$ μm²/s) (Fig. 6b). Interestingly, there is a strong correlation between phase length and diffusion coefficient, with the fastest diffusing particles spending the shortest time in that phase (Fig. 6c). Although some differences in diffusion coefficients were noted between experimental groups, we did not observe a meaningful trend (Supplementary Fig. 6b), and all plots of $D$ against phase length resulted in a similarly shaped distribution (Supplementary Fig. 6c), suggesting that diffusion rates are inherent to the mode of diffusion itself. Together, these data indicate that while XPA's entry into a paused state is dependent upon the disordered N-terminal and C-terminal domains, phase lengths and diffusion rates are dictated by the core DBD.

The calculated theoretical limit to the diffusion coefficient for Qdot-flXPA sliding and following the helical path ($D_{lim}$) is 0.015 μm²/s (Supplementary Note 3) and appears to separate the short-range from the long-range mode (Fig. 6b, c). The fact that a significant portion of the long-range phases exceed this limit suggests that, when in this mode, XPA is undergoing an alternative mechanism of linear diffusion, namely hopping. To reconcile this, we compared the behavior of flXPA on UVλ$_{20J}$ and UVλ$_{80J}$ tightropes in buffers with different ionic strengths (Supplementary Movie 3). Proteins that are hopping along the DNA are expected to exhibit faster diffusion in higher salt concentrations, while those that are truly sliding on DNA are expected to be relatively unaffected by changes in salt[57,58]. Comparing the maximum displacement of XPA, we observed a significant correlation between salt concentration and range of motion (Supplementary Fig. 7a). Diffusion coefficients for XPA undergoing short-range diffusion are not affected by added salt (Supplementary Fig. 7b), thus supporting the sliding model. However, consistent with the hopping model, increased salt concentration resulted in an increase in the diffusion coefficient for XPA undergoing long-range motion.

**Discussion**
In this study we have pursued a collection of single molecule experiments that uniquely address several questions about how

XPA interacts with DNA. AFM data indicate that full-length human XPA has specificity for the helix-bending dG-C8-AAF lesion embedded in a long DNA substrate and that it binds and bends both damaged and non-damaged DNA as a monomer. Single molecule fluorescence microscopy showed that XPA is primarily (~70%) stationary when bound to DNA. The motile particles exhibited episodic stalling and the presence of DNA damage increased pausing in a dose-dependent manner. A truncated XPA variant consisting of just the core DNA-binding domain (M98–T239) displayed impaired pausing compared to full-length XPA on damaged DNA tightropes. MSD analysis revealed that the long-range mode of linear diffusion had a significantly higher diffusion coefficient than the short-range mode.

Although XPA has been reported to bind DNA as a homodimer on short DNA substrates[9], volumes of XPA on a 538 bp DNA substrate measured by AFM in this study are consistent with the size of a monomer, in support of previous reports[15,21,36]. We also observed formation of a "dimer band" at high concentrations by EMSA, but this band is indistinguishable from a complex containing one XPA bound at the lesion and one bound at the DNA end (or two otherwise distinct monomeric binding events). Indeed, our AFM data show that a significant fraction of XPA was bound to one of either ends of the DNA molecule in addition to specific sites. Previous work has addressed the issue of end-binding by performing protein-protein crosslinking experiments of Rad14 DBD on 15 or 37 bp AAF-adducted DNA[10]. Dimers observed in this study may be a unique property of truncated Rad14 or due to the high protein concentrations required for such experiments (mM). As such, while our current data do not exclude the possibility that two proteins may bind damaged DNA at very high concentrations, AFM imaging of full-length human XPA on long DNA substrates has allowed us to gain new insight into XPA binding stoichiometry. The monomeric state of free XPA observed at 40 nM by AFM and up to 80 μM by MALS provides further evidence against dimer formation.

The addition of a dG-C8-AAF base modification at 30% from one end of our AFM DNA substrate resulted in increased flexibility in the DNA. Measurements of the DNA bend angle at 30% from each end of the molecule revealed two populations of angles centered ~10° and ~35°, consistent with previous reports that AAF distorts B-form DNA[49,50]. Further work will be required to confirm the nature of the 10° bend. XPA was shown to bend both damaged and non-damaged DNA by ~60°, comparable to the 70° DNA bend observed in the Rad14 structure[10]. Extensive reports in the literature suggest that XPA binds preferentially to a distorted DNA helix[13,25] relative to non-damaged dsDNA, and our current work supports the hypothesis that XPA interrogates DNA by bending and testing for flexibility or pre-bent structures. The energy required to bend DNA at an AAF site is expected to be less than at a non-damaged site[10,59], and thus XPA may preferentially form a stable complex more readily at this and other DNA lesions. Specificity for a bent DNA structure may be crucial for XPA to function in both damage verification, as well as its action as a scaffold protein, bound to NER intermediates. For example, XPA binds with TFIIH to bent DNA with a single-stranded overhang[21].

We observed preferential XPA binding to dG-C8-AAF with a specificity factor of 660 by AFM, which is in the range of reported specificities of *Taq* MutS for its substrates, a T-bulge and G:T mismatch (1660 and 300, respectively)[38]. XPA had a similar distribution of binding positions as we have previously observed for Rad4-Rad23 on a fluorescein-adducted substrate[42]. While the specificity factors determined by EMSA and AFM cannot be directly compared, in support of the AFM data, the specificity of XPA for dG-C8-AAF compared to non-damaged DNA by EMSA was ~85-fold. These levels of specificity support the "discrimination cascade" model for damage verification during

NER[7,25,26]. While the prevailing view has suggested that XPA is recruited after melting of DNA by TFIIH at the site of a DNA lesion, our AFM data indicate that XPA is able to detect and bind to an AAF lesion embedded in a 538 bp DNA molecule.

The role of linear diffusion in target search was explored further via the DNA tightrope assay. The episodic behavior of XPA appears to be a relatively unique property for DNA binding proteins, as few other single-particle tracking studies have yielded similar results, save for a recent report on the SA1 protein sliding on telomeric DNA tightropes[60]. Based on our tightrope and AFM data, it appears that monomeric XPA cycles through three distinct states on DNA: rapidly hopping over distances greater than 2 kbp (690 nm), slowly sliding over short (<2 kbp) ranges of DNA while interrogating and bending local regions, and forming high-affinity complexes with sharply bent DNA (Fig. 7). DNA with a high propensity for bending, such as at a dG-C8-AAF adduct[49,50], or 6-4PP[54], promotes formation of these stable complexes. Our data indicate that switching between modes occurs stochastically on the seconds time scale, and is controlled by the core DBD. The C-terminal end of the DBD, contained within the truncXPA construct, has high levels of predicted disorder (Supplementary Fig. 1a) but can fold into an extended alpha helix[21,33], suggesting possible conformational changes associated with binding modes. XPA's short-range motion is similar to the constrained diffusion we and others have observed for Rad4[42], XPC[61], and PARP1[56], where the recognition protein remains in close proximity to the target site while making room for subsequent proteins to assemble. Thus, short-range diffusion by XPA may facilitate damage verification during the transition between initial damage recognition during NER and XPA's scaffolding function.

The paused and diffusive states exhibited by motile XPA molecules are reminiscent of the two-state model proposed by Slutsky and Mirny[62]. In response to the speed-stability paradox of protein-DNA target search, the authors suggest that proteins adopt two conformations: a search state with a smooth DNA-binding energy landscape ($\sigma \lesssim 1-2 \times k_BT$), allowing for rapid search, and a recognition state with a rugged energy landscape ($\sigma \gtrsim 5 \times k_BT$)[62,63]. Linear diffusion observed for XPA fits this model. Paused particles conform to a recognition state while short-range and long-range diffusing particles correspond to distinct subgroups (and thus distinct energy landscapes) within a search state.

The energy barrier to free diffusion ($\sigma = 1.6 \times k_BT$, Supplementary Note 3) in the short-range mode (i.e., displacement less than 690 nm or ~2 kbp) is essentially the same as Rad4-Rad23 undergoing constrained motion[42]. We attribute the slower rate of diffusion observed in the short-range mode to two factors. First, as XPA bends DNA to assess damage/helical distortions, this likely induces propagation of the bend along the DNA molecule, thereby increasing the roughness of the energy landscape and ultimately limiting diffusion[64]. Second, XPA is translocating along the DNA by spiraling in constant contact with the helix.

Diffusion coefficients determined for XPA in the long-range mode, however, were affected by the ionic strength of the buffer and exceeded $D_{lim}$ for XPA sliding in this manner (and consequently, we are unable to calculate corresponding energy barriers to diffusion). Therefore, it is possible that XPA is hopping (i.e., rapidly undergoing micro-associations and dissociations from the DNA) in order to achieve the faster diffusion observed. In the cell, a search mechanism involving hopping would have the advantage of the protein being able to move along the DNA without being stopped by other protein-DNA complexes or nucleosomes.

Truncated XPA, consisting of residues 98–239 of the core DNA-binding domain, maintains its ability to adopt a recognition state on damaged DNA. However, motile particles of truncXPA were significantly impaired in their ability to pause on damaged DNA, as demonstrated by a decreased number of pause sites and

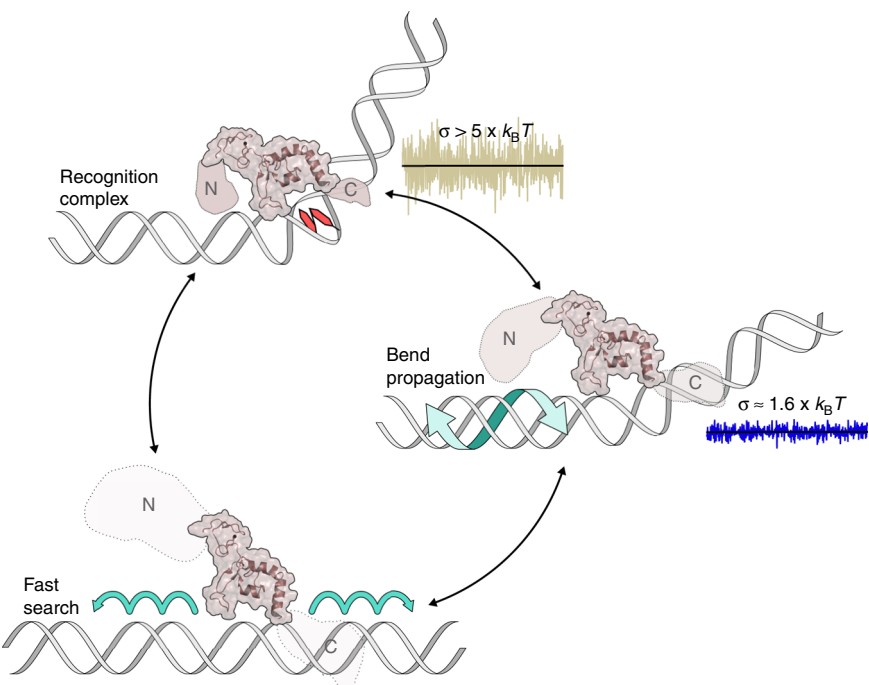

**Fig. 7 Working model of XPA linear diffusion on DNA.** Fast search (long-range linear diffusion): monomeric XPA translocates rapidly by hopping (micro-associations/dissociations) along the DNA backbone. The protein's disorder and lack of contacts with DNA permit rapid diffusion ($D \approx 0.04\ \mu m^2/s$). Bend propagation (short-range linear diffusion): XPA follows the path of the major or minor groove, spiraling along DNA ($D \approx 0.003\ \mu m^2/s$). The N- and C-termini of the protein may be partially folded, making more contacts with the DNA. The protein bends DNA, testing for flexibility/aberrations, and propagation of the bend further slows diffusion. Recognition complex (paused and stationary particles): the disordered ends of the protein fold into a stable complex, bending the DNA ~60°. The NMR structure of the XPA DNA-binding domain (PDB 1XPA) was used in combination with an estimation for the N- and C-termini (disorder represented by size/transparency) as an approximate model to display the complete XPA molecule.

a decreased total time spent in the paused mode. Compared to flXPA, truncXPA had approximately half the specificity for AAF-adducted DNA, and displayed no specificity for CPD-containing DNA. We therefore suggest that the N-terminal and C-terminal arms likely play a significant role in helping to identify alterations in the DNA helix and may serve as metaphorical brakes, allowing XPA to slow its search and test the site for a lesion or help in recruiting other repair factors. These data are consistent with a model in which the core DNA-binding domain is able to form stable complexes, but the disordered arms play a role in embracing the DNA and interrogating for helix-distorting damage, inducing a conformational change in both the protein and DNA to initiate pausing (Fig. 7). As NER proceeds and XPA plays a major role as a scaffold protein, the arms could then be available to interact with other DNA repair proteins. This is supported by published interactions of RPA32, ERCC1, and TFIIH (XPD) with the disordered N-terminus of XPA, as well as TFIIH (p52 and p8/TTDA) with the XPA C-terminus[21,23]. Given the many reported interactions between XPA and other NER proteins[23], XPA may show altered dynamics on DNA in the presence of other repair partners. Thus, we present our model as an important starting point, illustrating XPA-DNA binding behavior and diffusive properties in isolation.

## Methods

**Protein purification.** *His-flXPA.* Full-length, wild-type human XPA cDNA was cloned into the pIBA35 vector with an N-terminal His tag. The plasmid was transformed into One Shot BL21(DE3)pLysS competent *E. coli* cells (Invitrogen). Cultures were grown in LB medium containing 100 μg/ml ampicillin and 34 μg/ml chloramphenicol at 37 °C until the OD₆₀₀ reached ~0.6. At this point, expression was induced with 0.5 mM IPTG and 10 μM ZnCl₂ and cultures continued to grow for 4 h. Cells were harvested by centrifugation at $16,000 \times g$ for 8 min at 4 °C. All the following purification steps were performed on ice or at 4 °C. Cell pellets were resuspended in His-XPA lysis buffer (25 mM Tris-HCl, pH 7.5, 100 mM NaCl, 5

mM DTT, 10% (v/v) glycerol, 10 μM ZnCl₂, and 30 mM imidazole), lysed by sonication, and the insoluble fraction was pelleted by centrifugation at $45,000 \times g$ for 45 min. The supernatant was loaded onto an equilibrated HisTrap HP nickel column (GE) and washed with 30 column volumes (CV) of His buffer A (25 mM Tris-HCl, pH 7.5, 100 mM NaCl, 5 mM DTT, 10% glycerol, 10 μM ZnCl₂, and 30 mM imidazole). The sample was eluted with a gradient of 0–100% in 10 CV, His buffer A to His buffer B (25 mM Tris-HCl pH 7.5, 100 mM NaCl, 5 mM DTT, 10% glycerol, 10 μM ZnCl₂, and 500 mM imidazole). Fractions containing XPA were pooled and loaded onto an equilibrated MonoQ column and washed with 5 CV MonoQ buffer A (25 mM Tris-HCl, pH 7.5, 50 mM NaCl, 5 mM DTT, 10% glycerol, and 10 μM ZnCl₂). XPA was eluted with a gradient of 0-100% in 15 CV MonoQ buffer A to MonoQ buffer B (25 mM Tris-HCl, pH 7.5, 1 M NaCl, 5 mM DTT, 10% glycerol, and 10 μM ZnCl₂). Fractions containing XPA were pooled, diluted with MonoQ buffer A, loaded onto an equilibrated Heparin column, and washed with 5 CV MonoQ buffer A. XPA was eluted with a gradient of 0-100% in 35 CV MonoQ buffer A to MonoQ buffer B. Fractions containing XPA were pooled and loaded onto a size exclusion column (HiLoad 16/60 Superdex 200), which was equilibrated with the His-XPA SEC buffer (50 mM HEPES, pH 8.0, 150 mM NaCl, 5 mM MgCl₂, 5 mM DTT, 5% glycerol, and 10 μM ZnCl₂). XPA eluted as a single peak and peak fractions were pooled, aliquoted, and flash frozen in liquid nitrogen and stored at −80 °C.

*His-flXPA-StrepII and His-truncXPA-StrepII.* Full-length, wild-type human XPA cDNA was cloned into the pIBA43 vector with an N-terminal His tag and C-terminal StrepII tag. To make truncated XPA, the N- and C-termini were deleted from this plasmid (cloning by Gene Universal), leaving only residues 98–239. Both constructs were expressed and purified as described below.

The plasmid was transformed into One Shot BL21(DE3)pLysS competent *E. coli* cells (Invitrogen). Cultures were grown in LB medium containing 100 μg/ml ampicillin, 34 μg/ml chloramphenicol, and 10 μM ZnCl₂ at 37 °C until the OD₆₀₀ reached ~0.4. The temperature was then decreased to 16 °C and growth was continued until an OD₆₀₀ of ~0.6 was achieved. At this point, expression was induced with 0.2 mM IPTG and cultures continued to grow overnight. Cells were harvested by centrifugation at $5000 \times g$ for 30 min at 4 °C. All the following purification steps were performed on ice or at 4 °C and, between each step, samples were analyzed on 4–12% Bis-Tris SDS gels and stained with SimplyBlue SafeStain (Invitrogen). Cell pellets were resuspended in His-XPA-StrepII lysis buffer (50 mM potassium phosphate, pH 7.9, 200 mM KCl, 20 mM imidazole, 5% glycerol, 0.02% sodium azide, and EDTA-free Protease inhibitor cocktail), lysed by sonication, and the insoluble fraction was pelleted by ultracentrifugation at $147,000 \times g$ for 2 h. The supernatant was loaded onto an equilibrated HisTrap HP nickel column (GE) and

washed with 30 CV His buffer C (3.5 mM KH₂PO₄, 46.5 mM K₂HPO₄, 200 mM KCl, 20 mM imidazole, 5% glycerol, and 0.02% sodium azide). The sample was eluted with a gradient of 0–50% His buffer C to His buffer D (3.5 mM KH₂PO₄, 46.5 mM K₂HPO₄, 200 mM KCl, 500 mM imidazole, 5% glycerol, and 0.02% sodium azide). Fractions containing XPA were pooled and loaded onto an equilibrated StrepTrap HP column (GE) and washed with 50 column volumes Strep buffer A (50 mM Tris-HCl, pH 8.0, 200 mM KCl, and 0.02% sodium azide) and eluted with Strep buffer B (50 mM Tris-HCl, pH 8.0, 200 mM KCl, 5 mM desthiobiotin, and 0.02% sodium azide). Fractions containing XPA were pooled and dialyzed into His-XPA-StrepII SEC buffer (50 mM HEPES, pH 8.0, 200 mM KCl, 5 mM MgCl₂, 5 mM DTT, 10% glycerol, and 0.02% sodium azide). Size exclusion chromatography was performed using the AKTA FPLC with a HiLoad 16/60 Superdex 200 column (Amersham). XPA eluted as a single peak and peak fractions were pooled, aliquoted, and flash frozen in liquid nitrogen and stored at −80 °C. His-XPA-StrepII and His-XPA were compared by EMSA and AFM to confirm that the addition of the StrepII tag did not affect protein behavior.

His-flXPA was used in all AFM experiments and some DNA tightrope experiments. His-flXPA-StrepII was used for all electrophoretic mobility shift assays, multiangle light scattering, and some DNA tightrope experiments. To verify that both protein preparations exhibited similar behavior, they were compared by AFM with respect to the following parameters: binding position on AAF₅₃₈, induced DNA bend angle, and AFM volume (Supplementary Fig. 5a–d). Because His-flXPA-StrepII has a slightly higher molecular weight than His-flXPA, His-flXPA-StrepII results were only used for validation and were not combined with His-flXPA results for AFM experiments. Since both preparations behaved similarly in our AFM studies, we combined the data obtained in the DNA tightrope experiments.

**Multiangle light scattering.** Multiangle light scattering combined with size exclusion chromatography (SEC-MALS) was used to determine the oligomeric state of purified His-flXPA-StrepII in solution, as previously described[65,66]. Briefly, 100 μL of His-flXPA-StrepII at concentrations of 65 μM or 80 μM were injected on a Superdex 200 10/300 column (GE Healthcare) pre-equilibrated with XPA MALS buffer (50 mM HEPES, pH 8.0, 200 mM KCl, 5 mM MgCl₂, 5 mM DTT, and 10% glycerol) and eluted at a constant flow rate of 0.5 ml/min at room temperature. The molecular mass of the eluting sample was determined using the light scattering signal from a Dawn Heleos 8+ detector and the concentration signal from an Optilab T-rEX refractive index detector (both Wyatt Technologies). Data analysis was carried out with the ASTRA software (version 6.1.5.22, Wyatt Technologies).

**DNA substrate preparation.** *AAF₃₇ oligo.* The 37 nt oligonucleotide containing one dG-C8-AAF lesion (AAF₃₇-top, sequence below) was prepared as published[10]. Briefly, the synthesis was performed on an ABI 394 Nucleic Acid Synthesis System (Life Technologies) using ultra-mild conditions and commercially available phosphoramidites from Glen Research and Link Technologies. The AAF lesion was introduced with the help of a dG-AAF phosphoramidite with an iso-propylphenoxyacetyl protecting group at the N²-position[10]. The coupling time of the dG-AAF phosphoramidite was extended to 2 × 10 min to achieve good coupling yields. The crude oligonucleotide was deprotected using the reported ultra-mild conditions and purified by reverse phase high-performance liquid chromatography. Final desalting was performed with a Sep-PAK cartridge (Waters).

*37 bp DNA duplexes for EMSA.* Oligonucleotide sequences (G^AAF = dG-C8-AAF; T<>T = CPD; 6FAM = fluorescein; all purchased from IDT except AAF₃₇-top):

ND₃₇-top: 5'-phosphate-CCGAGTCATTCCTGCAGCGAGTCCATGGGAGT CAAAT

AAF₃₇-top: 5'-phosphate-CCGAGTCATTCCT**G^AAF**CAGCGAGTCCATGGG AGTCAAAT

CPD₃₇-top: 5'-phosphate-CCGAGTCATTCCTGCAGCGA**T<>T**CCATGGG AGTCAAAT

FAM₃₇-bottom: 5'-6FAM-ATTTGACTCCCATGGACTCGCTGCAGGAATG ACTCGG

CPD₃₇-bottom: 5'-6FAM- ATTTGACTCCCATGGAATCGCTGCAGGAAT GACTCGG

ND₃₇ was prepared by annealing ND₃₇-top and FAM₃₇-bottom. AAF₃₇ was prepared by annealing AAF₃₇-top and FAM₃₇-bottom. CPD₃₇ was prepared by annealing CPD₃₇-top and CPD₃₇-bottom. Annealing reactions contained 1.25 μM top strand, 1 μM bottom strand, 10 mM Tris-HCl, pH 8.0, and 100 mM KCl. Reactions were incubated at 95 °C for 5 min then cooled slowly to room temperature.

*Defined lesion plasmids.* Plasmids containing single site-specific dG-C8-AAF adducts were prepared as described previously[53,55]. Briefly, purified pSCW01 plasmids were nicked by Nt.BstNBI to create a 37-base gap. A 37mer containing a single dG-C8-AAF (AAF₃₇-top, above) was annealed into this gap and the backbone was sealed with T4 DNA ligase.

*DNA duplexes for AFM.* Substrates for AFM were prepared as described previously[55]. Essentially, a 538 bp DNA fragment was cut out of either unmodified pSCW01 plasmid (for ND₅₃₈) or pSCW01 with a site-specific dG-C8-AAF lesion, described above (for AAF₅₃₈). The plasmid was incubated with restriction enzymes XmnI and PciI, cutting 372 bp 5' to and 165 bp 3' to the lesion, respectively. Nick₅₁₄

was prepared by amplifying a 514 bp fragment from the pSCW01 plasmid and treating with Nt.BspQI to create a nick at 36% of the DNA contour length.

*Long DNA substrates for tightrope assay.* The NDλ substrate was prepared by diluting λ genomic DNA (NEB) to 50 ng/μl in 10 mM Tris-HCl, pH 8.5. The UVλ₂₀ⱼ substrate was prepared by treating NDλ with 20 J/m² of UV-C radiation (254 nm). A qPCR assay was performed previously to confirm the presence of UV photoproducts (6-4 PPs and CPDs) at a density of ~1 lesion per 2.2 kbp (ref. [53]). To prepare UVλ₈₀ⱼ and increase the lesion density such that there was ~1 lesion per 550 bp (i.e., 1 6-4PP every 2.2 kbp), NDλ was treated with 80 J/m² of UV-C. The dG-C8-AAF arrays were prepared as described previously[53,55], using the defined lesion plasmid described above. Lesion-containing pSCW01 was linearized via restriction digest by XhoI (NEB) then incubated with T4 DNA ligase (NEB) to achieve long (>40 kbp) tandemly ligated products with one dG-C8-AAF every 2 kbp.

**Electrophoretic mobility shift assay (EMSA).** XPA-DNA reactions were prepared by combining 8 nM 37 bp DNA with varying amounts of His-flXPA-StrepII in XPA EMSA buffer (20 mM HEPES, pH 7.5, 150 mM NaCl, 5 mM DTT, 0.5 mg/ml BSA, and 5% glycerol) in a final reaction volume of 10 μl. Each reaction was incubated for 25 min at room temperature then immediately loaded on two pre-run 5% non-denaturing polyacrylamide gels (37.5:1 acrylamide:bis). Both pre-run and run were performed at 4 °C, in 0.5× TBE buffer (44.5 mM Tris, 44.5 mM boric acid, 1 mM EDTA, pH 8.4), at constant voltage (90 V). DNA bands were visualized using a laser scanner for fluorescence (Typhoon, Amersham).

Gel images were quantified by measuring signal intensities of each band (ImageJ, NIH). The percentage of DNA bound was determined by dividing the intensity of the shifted ("bound") DNA by the sum of all bands in a lane. These values were plotted against XPA concentration and the data were fit to the following equation via nonlinear regression (GraphPad Prism):

$$\% \, \text{DNA Bound} = 100$$
$$\times \frac{(P + D + K_D) - \sqrt{(P + D + K_D)^2 - 4PD}}{2D} \tag{1}$$

where $K_D$ is the equilibrium dissociation constant, $P$ is the total protein concentration, and $D$ is the total DNA concentration. This model was chosen because our experimental conditions required that the DNA concentration be in the same molar range as the $K_D$ (ref. [67]).

**Atomic force microscopy (AFM).** *Sample preparation.* Samples for AFM were prepared as previously described[55]. All buffers and solutions were first filtered through 0.02 μm sterile filters (Whatman). For imaging of free proteins or free DNA (i.e., no reaction), the sample was diluted to either 40 nM (protein) or 4 nM (DNA) in AFM deposition buffer (25 mM HEPES, pH 7.5, 25 mM NaOAc, and 10 mM Mg(OAc)₂) which had been pre-warmed to 65 °C and brought back to room temperature. XPA-DNA reactions consisted of 100 nM 538 bp DNA (ND₅₃₈ or AAF₅₃₈) and 0.6–4 μM His-flXPA in XPA AFM buffer (50 mM HEPES, pH 8.0, 150 mM NaCl, 5 mM MgCl₂, 10 μM ZnCl₂, 5 mM DTT, and 5% glycerol) in a total volume of 10 μl. APE1-DNA and Polβ-DNA reactions consisted of 100 nM 514 bp DNA (Nick₅₁₄) and 500 nM protein in APE1 buffer (50 mM HEPES, pH 7.5, 150 mM NaCl, 10 mM MgCl₂) in a total volume of 10 μl. Each binding reaction was incubated for 30 min at room temperature then diluted 1:25 in AFM deposition buffer. Twenty five microliter droplets were deposited on freshly cleaved mica, allowed to equilibrate for 30 s with gentle rocking, then washed with 1 ml of filtered H₂O and dried under a gentle stream of nitrogen gas.

*Data collection.* All AFM images were obtained using ScanAsyst PeakForce Tapping mode in air on a Multimode V Microscope with an E scanner (Bruker). Samples were scanned with a triangular tip with a nominal radius of 2 nm, mounted on a silicon nitride cantilever (SCANASYST-AIR, Bruker). Probes were replaced for each new experiment or more frequently as needed. 1 × 1 micron images were collected at a resolution of 512 × 512 pixels and a scan rate of 0.977 Hz. The peak force setpoint was 0.01988 V.

*Data analysis: free protein standard.* To generate the standard curve relating AFM volumes to molecular weight, analysis was performed on AFM images with the isolated protein samples. The following proteins of known MW were used: recombinant human HMGB1 (Abcam), His-tagged human APE1 (gift from Sam Wilson), His-tagged human DNA polymerase β (gift from Sam Wilson), and His-tagged UvrD (purified as published[68]). Particle dimensions were measured using Image SXM software and used to calculate volumes:

$$V = A \times (H - B) \tag{2}$$

where $V$ is the particle volume, $A$ is the area of the particle footprint (determined via a set density threshold above background noise), $H$ is the mean height of the particle, and $B$ is the background height of the overall image[69].

*Data analysis: intrinsic DNA bend angle.* To determine the intrinsic bend angles of DNA substrates at 30% from each end, AFM images containing only the DNA were analyzed. DNA molecules used in this analysis had to be completely visible and isolated (i.e., not continuing past the edge of the image nor overlapping with itself or another molecule) and the total contour length must be within the range of ±10% of the expected length. Measurements were done on TIF images using

ImageJ software (NIH). The total DNA contour length was first measured and points at 30% from both ends were marked. Local bend angles at these sites were measured and are reported as the supplementary angle, θ (Fig. 4).

*Data analysis: protein-DNA complexes.* In addition to the criteria for usable DNA molecules as described above, the analysis of protein-DNA complexes first required the identification of bound proteins using the following criteria: (a) the height of the complex must be greater than the average height of the DNA molecule and (b) the complex width must be greater than the average width of the DNA molecule.

Methods for measuring protein binding position and induced DNA bend angle using ImageJ software (NIH) have been described in detail[55]. Briefly, binding position was determined by dividing the contour length of the DNA molecule from the center of a bound protein to the closest DNA end by the total DNA contour length. XPA-induced DNA bend angles were measured at sites of bound XPA. In cases where two or more proteins were bound to the same DNA molecule, angles were not measured.

To measure protein volume when bound to DNA, the DNA volume was estimated and subtracted from the total complex volume (Supplementary Fig. 4A). Image SXM software was used to trace the perimeter of the complex. The length of the DNA through this space was projected assuming that the DNA runs through the center of the complex. Then, two unbound regions of DNA on either side of the complex, with lengths corresponding to that of the complex, were delineated. In cases where the protein was bound near the end of the DNA or near another protein, two unbound regions of DNA were chosen at other available locations on the same molecule. Volumes of all three regions (complex, DNA1, and DNA2) were determined as described above (Equation 2). Protein volume was determined as the total complex volume minus the average of the two unbound DNA volumes:

$$V_{protein} = V_{complex} - \frac{V_{DNA1} + V_{DNA2}}{2} \tag{3}$$

Histograms of all AFM results were plotted and Gaussians were fit to the data by nonlinear regression in GraphPad Prism. The number of histogram bins corresponds to the square root of the sample size.

**DNA tightrope assay**. *Flow cell set-up.* All steps for reagent/material preparation, flow cell set-up, protein labeling, imaging, and data analysis for the tightrope assay have been described in detail[55] according to methods developed previously[51,52]. Briefly, flow cells were prepared by attaching slides with inlet/outlet tubing to PEGylated coverslips via tape spacers. Flow cells were incubated in blocking buffer (10 mM HEPES, pH 7.5, 50 mM NaCl, and 1 mg/ml BSA) for 10 min, then poly-L-lysine coated silica microspheres (5 μm diameter) were flowed in and dispersed across the coverslip. Long DNA substrates were suspended between beads using continuous hydrodynamic flow with alternating direction in tightrope buffer (20 mM HEPES, pH 7.5, 50 mM NaCl, 3 mM MgCl₂).

*Protein labeling.* His-flXPA, His-flXPA-StrepII, or His-truncXPA-StrepII was labeled with either 705 nm or 605 nm quantum dots (Qdots). The former strategy was accomplished by first incubating streptavidin-coated 705 Qdot (Invitrogen) with biotinylated anti-His antibody, at a final concentration of 167 nM Qdot and 833 nM antibody. Then, this mixture was incubated with an equal volume of 167 nM XPA; the final XPA concentration was 83.3 nM. The latter strategy was accomplished by first incubating XPA with a mouse monoclonal anti-His antibody, both at a final concentration of 200 nM. Then, this mixture was incubated with an equal volume of 1 μM 605 Qdot conjugated to an anti-mouse secondary antibody (Invitrogen); final XPA concentration was 100 nM. Labeled protein mixtures were diluted 4:100 in XPA tightrope buffer (25 mM HEPES, pH 8.3, 100 mM KCl, 1 mM EDTA, 0.55 mg/ml BSA, 1 mM DTT, and 10% glycerol). The final XPA concentration in the flow cell was 3–4 nM. To check for background binding by the Qdots or antibodies, controls were performed using the above conjugations with buffer instead of XPA. To verify that the two different Qdot labeling strategies did not impact results, the behavior of XPA labeled with either the 605 Qdot or 705 Qdot was compared on UVλ₂₀ⱼ tightropes, showing no significant difference (Supplementary Fig. 5e).

For 150 mM NaCl experiments, 150 mM NaCl buffer (25 mM HEPES, pH 7.5, 150 mM NaCl, 1 mM DTT, and 0.1 mg/ml BSA) was used in place of XPA tightrope buffer. For 1 M NaCl experiments, stationary particles of XPA were recorded in XPA tightrope buffer and, during recording, the buffer in the flow cell was replaced with 1 M NaCl buffer (1 M NaCl, 20 mM HEPES, pH 8.3, 80 mM KCl, 0.8 mM EDTA, and 8% glycerol), taking ~40 s of flow.

*Data collection.* Movies of XPA-DNA interactions were recorded on an inverted fluorescence microscope (Nikon Ti) with an 100X oil-based high-NA objective for TIRF-M and a high-speed sCMOS camera (Andor and Teledyne Photometrics). Qdots were excited with a 488 nm laser at an optimal oblique (sub-TIRF) angle and visualized without an emission filter. Movies were taken for 5 min with frame rates between ~10 and ~12.5 fps.

*Data analysis.* Movie files were converted to a time series of individual TIF files (NIS-Elements, Nikon) and imported into ImageJ (NIH). Kymographs were generated using the slice function over the trajectory of the particle along the DNA. These were processed by FFT bandpass filtering to reduce noise (filter range 3–40 pixels, with suppression of vertical stripes). A Gaussian Fit plugin was used to fit the fluorescence intensity in the kymograph to a one-dimensional Gaussian at each point along the x-axis (i.e., each frame or time point)[51]. Fitting

data was processed using custom scripts in MATLAB (MathWorks) to exclude poorly fitted positions and convert particle position from pixels to nm.

First, each particle (i.e., one kymograph, 5 min observation) was categorized based on whether it moved at all during the observation window (stationary vs. motile) and whether it dissociated during recording (persistent vs. dissociated). Dissociation was defined as the disappearance of Qdot-XPA for at least 200 frames (~20 s). We have previously reported that the mean positional accuracy for a 605 nm Qdot bound to biotin is $6 \pm 3$ nm by Gaussian fitting of the fluorescence intensity to a point spread function[53]. The position uncertainty over time has been determined to be $36 \pm 3$ nm (~100 bp), accounting for stage drift, DNA movement, and thermal fluctuations[53]. We used a conservative cutoff of 130 nm (three pixels, ~400 bp) to classify motile particles[42,56].

Motile particles were analyzed further for different modes of diffusion. Each kymograph was broken down into phases, falling into three possible modes of behavior: paused (particle displacement not varying more than 130 nm), short-range diffusion (displacement between 130 nm and 690 nm), and long-range diffusion (displacement greater than 690 nm). Shorter range modes were only allowed to interrupt longer range modes if they persisted for at least 5 s. For example, if a particle was exhibiting short-range behavior, paused (i.e., stationary) for 10 s, then went back to short-range behavior, this would be counted as three phases; if the pause only lasted 2 s, this would be counted as a single short-range phase. If a particle was paused prior to recording, the first phase of a kymograph may be less than 5 s. Each motile particle was analyzed with respect to the following parameters: position range, phase switch rate, lifetime of each phase, and number of pause sites.

The mean squared displacement (MSD) was calculated for all motile phases (short-range and long-range) using custom scripts in MATLAB:

$$MSD(n\Delta t) = \frac{1}{N-n} \sum_{i=1}^{N-n} (x_{i+n} - x_i)^2 \tag{4}$$

where $N$ is total number of frames in the phase, $n$ is the number of frames at a given time step, $\Delta t$ is the time increment of one frame, and $x_i$ is the particle position in the $i$th frame[70]. The diffusion coefficient ($D$) was determined by fitting a linear model of one-dimensional diffusion to the MSD plots:

$$MSD(n\Delta t) = 2D(n\Delta t) + y \tag{5}$$

where $y$ is a constant ($y$-intercept). Fittings resulting in $R^2$ less than 0.8 or using less than 10% of the MSD plot were not considered.

**Reporting summary**. Further information on research design is available in the Nature Research Reporting Summary linked to this article.

## Data availability
All data are available from the corresponding author upon reasonable request.

## Code availability
Custom MATLAB scripts are available from the corresponding author upon reasonable request.

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

## Acknowledgements

The authors wish to thank Samuel Wilson for providing purified His-Polβ and His-APE1 and Lili Liu for providing Nick$_{514}$ and purified His-UvrD. The authors also thank Muwen Kong for helpful discussions and training, Neil Kad and Hong Wang for helpful discussions, and Namrata Kumar for careful reading of the manuscript. This work was supported by the National Institutes of Health [R01ES019566 to B.V.H., R01ES028686 to B.V.H., T32GM088119 to E.C.B., and 2P30CA047904 to UPMC Hillman Cancer Center]; Deutsche Forschungsgemeinschaft [SFB-1361, TP2-Carell].

## Author contributions

E.C.B. and B.V.H. conceived the research. E.C.B. and S.J. performed all DNA tightrope experiments. E.C.B. performed all AFM experiments. E.C.B. and I.C.D. analyzed all single-molecule data. E.C.B. and S.J. performed and analyzed all EMSA experiments. J.K., C.K. and F.S. performed and analyzed SEC-MALS experiments. E.C.B., S.J., J.K., J.B., N.S., and T.C. purified protein and DNA substrates. S.C.W. provided imaging resources. E.C.B. and B.V.H. drafted the paper, which was reviewed, discussed, and edited by all authors.

## Competing interests

The authors declare no competing interests.
