## [Peer Review File · Nature Communications]

Reviewers' comments:

Reviewer #1 (Remarks to the Author):

The manuscript discusses the results a set of very insightful single molecule experiments, namely AFM and single molecule fluorescence microscopy, designed specifically to characterize the interaction between the NER protein XPA and DNA, with and without one single dG-C8-AFF lesion. It is well structured and well written and the results obtained are very important in clarifying exhaustively a few important aspects still very much in debate in regards to XPA binding to DNA and its potential role in the NER pathway, namely:

- a) XPA binds both damaged and undamaged DNA as a monomer
- b) XPA is a monomer when unbound in solution
- c) XPA causes severe bending in the DNA helix upon binding
- d) XPA has a higher affinity for damaged sites

The latter point is well justified by the authors (although only qualitatively), in terms of the energetic cost of bending an already bent structure (because of the presence of the lesion), relative to the cost of bending a relatively straight one to the same degree. The higher affinity for bent sites can also explain the ability of XPA to locate preferentially at the damage site compared to the non-sequence specific binding observed for the ND538, and shown in Figure 1, panels D vs. E.

The only aspect that I do not find particularly convincing in this version of the manuscript is the suggested role of XPA in DNA damage detection/recognition. The authors themselves state in the introduction that early literature suggests that XPA is involved in the damage detection steps, but that more recent studies suggest that its primary role is as a scaffolding protein supporting the stability of the DNA repair bubble and coordinating the NER proteins/enzymes around it. Nevertheless, without any further discussion, the authors state that one of the main aims of the work is indeed to understand the role of XPA in damage detection, which seems a bit of a non-sequitur to me. In this regard, it would be important for the authors to present/discuss the evidence that they considered as a standpoint or that convinced them that XPA may indeed have a role in DNA damage detection.

Within this framework, it is not clear to me how once the XPA detected the damage the complex would evolve to support XPA's role as a scaffolding protein, migrating away from the damage site (pre-excision) and locating itself on one of the edges of the DNA repair bubble (see the comment Shell and Chazin, *Structure* (2012) 20(4):566 and refs therein). I believe that the authors should discuss their results within the context of the DNA excision step to clarify how their theory supports or clashes with the current view of the mechanism.

One point that may be of interest for such discussion is that XPA higher affinity for a "bent" DNA helix doesn't necessarily translate in a lesion recognition role. Indeed, within the current (and widely/generally accepted) view of the NER excision complex, XPA binds at one of the two junctions, either 3' or 5', not yet 100% proven which one, there are arguments/data supporting each. From a structural point of view looking at the two single strands opening in opposite directions, the junction is a bent DNA construct, not too different from a bent DNA double helix if the protein binds only one of the strands. Within this framework, it is possible that XPA has a higher affinity for bent DNA constructs, because it needs to locate at a junction and not because it is involved in the damage detection step. Maybe the authors should comment also on this point.

In summary, the topics of discussion I suggested above represent a minor revision that may require a slight change in this specific direction/aim. Nevertheless, I believe that such additional discussion would add to an already outstanding work and valuable contribution to the field. For this reason, with these minor revisions addressed, I strongly support the publication of the manuscript

in Nature Communications.

Minor point:

Reference from Kokic et al, bioRxiv page 3 line 75 should be listed with all other references for consistency.

Reviewer #2 (Remarks to the Author):

This manuscript reports on the use of single molecule studies of XPA binding to damaged and undamaged DNA. The authors phrase their study in the context of nucleotide excision repair (NER), where XPA is known to function. However, this is a purely biophysical study of the protein in isolation, so the results only remotely relate to the role of XPA in NER. The data confirm previously published work of high affinity binding to an AAF lesion, that XPA binds DNA substrates as a monomer, and that DNA is bent by $\sim 60^\circ$ at sites where XPA is bound. Many different modes of linear diffusion of XPA on damaged and undamaged DNA, and the authors place these results in a model of XPA functioning in the search for distorted duplex DNA associated with damage.

The manuscript is written in clear language with a good logical flow. The experimental design, data acquisition, and analyses are technically sound and appear to be of high caliber. However, there is a fundamental flaw in this study when it comes to transferring the observations of the effects of isolated XPA to the broader landscape of the NER pathway. It is well accepted (and even summarized in the Introduction) that XPA is not present at sites of DNA damage until the damage is recognized by other NER proteins and the TFIIH complex is recruited. TFIIH opens the damaged duplex and unwinds it around the site of damage, then XPA is recruited along with RPA. At this point, XPA is not freely diffusing in solution searching for damage in a duplex, hence it is completely unclear how the results pertain to XPA function in NER.

The results do show that the presence of DNA damage affects how isolated XPA binds and diffuses on the unencumbered substrate. But the absence of the many other components of the large NER machinery (over 30 different proteins) limits the scope and significance of their findings. Given these limitations, it is misleading to direct the Introduction to NER and to try and relate the data to NER, and then only in the last sentence state that the model presented is a 'starting point' and that XPA may well behave quite differently when bound to other proteins.

Overall, the results obtained are relevant to biophysical investigations of DNA repair, but only remotely to the role of XPA in NER. Potential functions of XPA outside of NER have been reported recently. Do one or more of these functional roles better relate to these data? As presented here, the significance and potential impact of these results make this manuscript better suited to a more specialized journal.

Concerns

1. The Introduction (pg 3, lines 57-62) states: "XPA has been reported to display specificity for UV-irradiated DNA 6-4PPs, N-(2'-deoxyguanosin-8-yl)-2-acetylaminofluorene (dG-C8-AAF), cisplatin, benzo[a]pyrene, non-hybridized bases, and other artificially distorted substrates. Additional studies revealed that XPA also binds preferentially to partially single stranded DNA and forked substrates more closely resembling unwound NER intermediates. It has been previously proposed that XPA binding to damaged DNA induces conformational changes in the protein that are associated with different binding modes." The experimental design and presentation of results in this manuscript are not consistent with the well accepted role of XPA as a scaffold organizing the many partner proteins in NER complexes through protein-protein interactions.

2. The findings on the stoichiometry of XPA binding to DNA are not new; it is broadly accepted that XPA does not dimerize. There are many studies supporting this, including the structure of truncated Rad14 protein, in which two molecules are bound to the damaged DNA duplex, but there are no direct protein-protein contacts between them.

3. The experimental design is lacking one important control: a DNA substrate with a static bend but no DNA damage (introduced for example by the uneven lengths of strands). This would inform the role of XPA in DNA damage recognition. On pg 11 (line 268) the heading says Presence of UV damage slows the rate of mode switching, leading to longer phases. It is difficult to assess if the differences observed are caused truly by the helix-distorting DNA damage or by the DNA distortion on its own attracting XPA to the substrate. While the authors discuss this possibility on pg 15, line 350, and it is consistent with lack of binding to a static bent DNA reported previously by Yang and colleagues (Yang et al., 2006), an appropriate control is required. This is especially important in the context of NER, as XPA may preferentially bind DNA that has already been bent by other components of NER complexes.

4. Pg 5, line 119 – Of all the complexes observed, 33% were bound to the ends of the DNA substrate. This is quite a high rate of non-specific binding. Even 22% for the AAF538 substrate seems to be quite a high ratio of non-specific to specific binding. Why? Some comment on this is required in the text. The occurrence of cases with two molecules bound at both ends also observed also need to be reported.

5. Pg 11, line 248 – Results show that most (60-70%) of the XPA protein is in a stationary mode on tightrope substrates. Is the remaining 30-40% fraction of motile protein sufficient to conclude that XPA actively screens for DNA-helix distortion? Does this ratio change when more than one damage site per DNA substrate is introduced? How do the authors explain that the motile fraction of the protein changes modes, including pausing, but the stationary fraction does not?

6. Is there any speculation on what the pausing positions on the tightropes are?

7. Pg 16, lines 391-399 – The authors speculate without any specific support that the disordered N- and C-termini may fold in and interrogate the DNA for lesions via DNA bending. Considering that XPA is known to function as the key scaffold for NER pre-incision complexes and the many studies showing that it interacts with many other NER proteins, it is illogical to speculate that the disordered N- and C-termini of XPA are important for DNA binding only. This speculation should be supported by data or completely deleted from the text.

Additional points.

1. Pg 3, line 68 – A zinc finger is a specific structural motif that is characterized by the coordination of one or more zinc ions in order to stabilize the fold; the term should be reserved for such motifs. XPA possesses a Zn-binding motif, but not a Zinc finger and the wording should be adjusted accordingly.

2. Pg 3, lines 74-75 – The Kokic et al. manuscript is now published in Nature Comm.

3. Pg 4, lines 99-100 – How do the KD values compare to previously published data?

4. Pg 6, line 150 – The expected molecular mass of XPA monomer is 32.6 kDa but on pg 3, line 66 XPA is described as a protein of molecular mass of 31.4 kDa. While the origin of this inconsistency (XPA-His vs XPA) is mentioned at the Figure 2 label, it should be also clarified in the main text.

5. Pg 7, line 164 – Wording should be changed for Because XPA is a small protein (...). For example, "XPA is a relatively small protein for precise AFM measurement (...)"

6. Pg 23, line 601 – XPA is known to bind a zinc ion, and the presence of Zn has an influence of the structure. Hence, use of EDTA in the buffer may affect the protein's fold, stability, and solubility. Were controls performed?

Reviewer #3 (Remarks to the Author):

The manuscript by Beckwitt et al. presents single molecule atomic force microscopy (AFM) and fluorescence studies of the nucleotide excision repair (NER) factor Xeroderma pigmentosum group A (XPA). The authors aim at characterizing conformational and dynamic properties of XPA-DNA complexes during lesion search and at a target lesion to understand the mechanism of lesion recognition by XPA.

XPA is an essential protein and mutations in XPA severely affect human health. XPA functions in the NER pathway that repairs a plethora of structurally and chemically diverse DNA lesions, including acetylaminofluorene (AAF), which is used in the study, and in particular and famously UV induced damages in the DNA. The exact role of XPA in the NER mechanism is still controversial. In addition, the full length protein in complex with DNA has not been accessible to X-ray crystallography due to the intrinsic disorder in parts of the protein. The questions addressed by the authors are therefore highly relevant and interesting. In their studies, the authors investigate XPA in isolation (without other NER components present), while in vivo XPA functions in NER in complex with the transcription factor IIH (TFIIH) helicase machinery. However, understanding the conformational dynamics of DNA interactions by the isolated XPA protein adds to our better appreciation of the role of XPA in NER lesion processing.

The authors strive to resolve the question of the oligomeric state of XPA, the DNA bending introduced by XPA in non-damaged DNA and at an AAF lesion, and the mode(s) of translocation on DNA in the absence and presence of lesions. Regarding the oligomeric state of XPA, contradictory findings had been reported in the past. Beckwitt and colleagues clearly establish the monomeric form of XPA to be the dominant species both in solution and when bound to DNA, using AFM and multi angle light scattering. AFM characterization of DNA bending by XPA revealed comparable bending at non-damaged and lesion sites, and the bending angle was similar to that observed in crystal structures (of the shortened construct). The authors argue with energetic cost of DNA bending as a means for XPA to distinguish between lesion and non-damaged DNA sites based on their different flexibilities, as has been proposed before by others for XPA and for other DNA repair proteins. Regarding XPA lesion search dynamics, the authors report several different modes of translocation on DNA and the frequent interchange between these different modes. They interpret the modes as XPA testing for damage (pausing), XPA carrying out one-dimensional diffusion on the DNA in its lesion search (short range motion), and XPA in a fast scanning mode (mid range or long range motion). In a model they summarize their conclusions that XPA lesion search involves three different states: XPA searches for lesions in a fast scanning complex with diffusion coefficient dependent on ionic strength of the buffer (which they interpret as an indication for hopping); XPA probes for lesions in a short range scanning complex with lower diffusion constants, which may be in part caused by DNA bending by the protein to probe for lesions; and XPA forms a strongly bent stable recognition complex at the lesion.

Overall, I find the topic and results of this study interesting and highly relevant and I mostly agree with their conclusions. I do however, have a few points of criticism listed below. After minor changes, I hence recommend the manuscript for publication in Nature Communications.

Specific points:

1. AAF binding specificity determination by AFM:

The distribution of XPA positions on the DNA does not really resemble a > 600-fold preference for

lesion binding (when compared, for example, to distributions in Yang et al. NAR 2005 as cited by the authors in this context). I agree with the authors that there is specificity for AAF by XPA, but I think the value is overestimated due to underestimation of non-specific background binding. Differently expressed: I think the Gaussian width in the fit to the distribution is too broad, artificially increasing the specific area that results in the high specificity value. A smaller binning interval might more clearly show whether the maximum at the lesion position is truly this broad or whether the Gaussian shaped peak is in fact much narrower, but would likely require more data points. In fact, the specificity that this distribution reflects likely fits nicely with that obtained by the authors from their EMSA analyses. However, based on the high flexibility at an AAF lesion, it actually seems surprising that the specificity of XPA for this lesion should not be larger than observed here, if energetic preference of enhanced flexibility sites determinesthe probability of occupancy. Furthermore, enhancement of TFIIH lesion recognition through XPA, as suggested previously (and also mentioned by the authors) is difficult to imagine with the low to moderate recognition specificity compared to – for instance, that shown for the XPD helicase of TFIIH (admittedly for different lesions). These points could be addressed.

2. XPA volume analyses by AFM:

While I appreciate the thoroughness of their volume calibration for their peak force AFM, I think this section could mostly be moved to the Supplement. It is obviously indispensable, but does not add to the manuscript story. The authors analyzed the volumes of two proteins, APE1 and Pol(beta) bound to DNA, to test the validity of their volume calibration also for DNA bound protein volumes. In this context, I also think that the originally measured volumes for APE1 and Pol(beta) (before DNA volume subtraction) ought to be given in the text, in addition to the final volumes after DNA subtraction. After the thorough consideration of the contribution from DNA binding mode by APE1 and Pol(beta) to overall measured volume, it would recommend itself to also discuss this for the protein of interest in this study, XPA to support the approach of subtracting the DNA volume to determine the molecular mass of the DNA bound complex. Again, I think the originally measured volumes (prior to DNA volume subtraction) should also be given, as well as the DNA volumes.

3. DNA bend angle analyses by AFM:

In the bend angle distributions for DNA in the absence of protein (Figures 4C and D), the first interval seems to be only half the size of all other intervals since it is centered at zero degrees bending. If this is the case, then at least in the histogram shown in (C), correction to full interval size for the first range of bend angles (0 to 5 or 10 or whatever degrees) would likely result in a distribution with exponential decay from 0 degrees (rather than the 5.6 degrees maximum reported here). For the distribution in (D), at the AAF position, the first maximum very likely remains at the second bin, ie. larger than 0 degrees. In addition, for a better comparison between the two distributions, ideally the interval (bin) size should be the same, which would then be limited by the sample with the smallest number of datapoints (also for Figures 4 G and H). Also, for smaller bin sizes, the distributions in Fig. 4 G and H may well reveal more than one bend angle species, as also suggested by the example images (but smaller binning will be difficult due to small samples sizes). Two different conformational states in lesion search and recognition by XPA would seem interesting also in the context of the final model proposed by the authors. However, none of this will drastically change the story.

4. DNA tightrope assay:

- a. The authors state that the proportion of immobile XPA is not affected by the DNA substrate. However, Figure 5B shows ~90% versus ~70% immobile XPA complexes on DNA (including dissociating and persistent molecules) for AAF versus UV or non-damaged substrate. The statistical non-significance level seems surprising! The authors state that the density of UV lesions in their substrate can be expected to be comparable to that in the AAF substrate (approximately one lesion per ~2,000 bp), yet the percentage of immobile complexes appears to differ between the UV and AAF substrates. Can the type of lesion explain the different behavior?
- b. The lack of medium/long range scanning species for AAF versus non-damaged or UV substrate

is striking. Long/medium range motion is defined by the authors as movement over > 690 nm, the distance between two AAF lesions and approximate average distance between two UV lesions assuming the above stated density. Exclusively short range scanning by XPA is hence consistent with XPA not proceeding over an AAF lesion. Again, could this depend on the type of lesion, since for the UV substrate a significant population of medium/long range scanning molecules was observed? In contrast to AAF, CPD lesions, as one type of UV lesion, do not strongly distort DNA.

c. Many of the sub-species do not seem to contain enough data points for statistical analyses. For example $n=3$ for long range diffusion coefficients of pooled UV and non-damaged substrates, or $n=3$ for maximum displacement at 1 M salt concentration. I appreciate that data collection with this type of single molecule experiment is time consuming, but would be careful to draw conclusions from such small sample sizes.

d. I do not agree with the classification of long range scanning particles based on the example trace shown in Figure 5D. In this example, the "long range" trace seems to consist simply of two medium range movements interrupted by a clear pausing plateau. I would suggest to pool the data from medium range and long range modes. In the population distributions (e.g. Figure 6F) the long range species never seems to truly add any additional information. Also, the issue of not enough data points for some of the different species (point above) may automatically be removed when pooling medium and long range scanning particles.

e. I also wonder if the salt concentration effect on the medium/long range but not on the short range moving species really supports hopping as the mode of translocation in the medium/long range species and one dimensional diffusion for the short range species. Regarding the short range species, the distribution of short range diffusion coefficients (Figure 6E) appears to indicate one species consistent with one dimensional diffusion (below the limit line for diffusion) and a second with higher diffusion constants (above the limit line for diffusion). Regarding the medium/long range species, it seems to me that hopping is only one possible explanation of the effect observed at higher salt concentration. Could the salt concentration not instead simply affect stability of DNA interactions by the protein? These interactions seem to be majorly electrostatic when looking at published structural data. The example in Supplementary Figure 5c showing transition from paused to long range diffusion when increasing the salt concentration to 1M would be consistent with this.

f. I wonder if DNA bending as seen in AFM applies for the tightrope assay or if the DNA is "fully" stretched between the anchoring beads? If XPA would not be able to bend the DNA as part of its lesion probing, this would affect the time spent probing for lesions (ie. bending the DNA, ie. in the paused state as well as in the short range linear diffusion state). The authors mention that other proteins previously tested in this assay did not show the changing between translocation modes seen for XPA. Were any of these proteins expected to bend the DNA?

Responses to each reviewer's comments

We greatly appreciate the care and depth of analysis that each reviewer has provided in their evaluation of our study.

In response to the reviewers' excellent points we have made major revisions to the manuscript (described in detail below, and marked in blue text in the manuscript). These included restructuring the results, refocusing the introduction and discussion per their concerns, doing extensive data re-analysis, and performing two new experiments (increased lesion density on the DNA tightropes and work-up of a truncated XPA variant), which have been incorporated into a new Figure 5. The study has been enhanced by the review process.

Please note all our responses

Reviewer #1

The manuscript discusses the results a set of very insightful single molecule experiments, namely AFM and single molecule fluorescence microscopy, designed specifically to characterize the interaction between the NER protein XPA and DNA, with and without one single dG-C8-AFF lesion. It is well structured and well written and the results obtained are very important in clarifying exhaustively a few important aspects still very much in debate in regards to XPA binding to DNA and its potential role in the NER pathway, namely:

- a) XPA binds both damaged and undamaged DNA as a monomer
- b) XPA is a monomer when unbound in solution
- c) XPA causes severe bending in the DNA helix upon binding
- d) XPA has a higher affinity for damaged sites

The latter point is well justified by the authors (although only qualitatively), in terms of the energetic cost of bending an already bent structure (because of the presence of the lesion), relative to the cost of bending a relatively straight one to the same degree. The higher affinity for bent sites can also explain the ability of XPA to locate preferentially at the damage site compared to the non-sequence specific binding observed for the ND538, and shown in Figure 1, panels D vs. E.

The only aspect that I do not find particularly convincing in this version of the manuscript is the suggested role of XPA in DNA damage detection/recognition. The authors themselves state in the introduction that early literature suggests that XPA is involved in the damage detection steps, but that more recent studies suggest that its primary role is as a scaffolding protein supporting the stability of the DNA repair bubble and coordinating the NER proteins/enzymes around it. Nevertheless, without any further discussion, the authors state that one of the main aims of the work is indeed to understand the role of XPA in damage detection, which seems a bit of a non-sequitur to me. In this regard, it would be important for the authors to present/discuss the evidence that they considered as a standpoint or that convinced them that XPA may indeed have a role in DNA damage detection.

The reviewer (as well as Reviewer #2) brings the prevailing view in the literature, that XPA is involved in later steps in NER. It is interesting to note that reviewer 3 indicates: "**The exact role**

of XPA in the NER mechanism is still controversial. In addition, the full length protein in complex with DNA has not been accessible to X-ray crystallography due to the intrinsic disorder in parts of the protein. The questions addressed by the authors are therefore highly relevant and interesting...understanding the conformational dynamics of DNA interactions by the isolated XPA protein adds to our better appreciation of the role of XPA in NER lesion processing.”

We also embrace this philosophy, and it was thus our intent to better understand XPA's interaction with DNA which we outline on the bottom of page 3 (lines 68-71). The introduction and discussion have been revised to emphasize the traditional role of XPA as a scaffold protein during NER. As attached at the end of the rebuttal (pg. 13) we have created a table showing that XPA displays specificity for specific DNA lesions and specific structures. We believe that the literature strongly indicates that XPA has specificity for bent structures, including largely distorting DNA lesions, which our new AFM data strongly support.

Within this framework, it is not clear to me how once the XPA detected the damage the complex would evolve to support XPA's role as a scaffolding protein, migrating away from the damage site (pre-excision) and locating itself on one of the edges of the DNA repair bubble (see the comment Shell and Chazin, Structure (2012) 20(4):566 and refs therein). I believe that the authors should discuss their results within the context of the DNA excision step to clarify how their theory supports or clashes with the current view of the mechanism.

We appreciate the reviewer raising this point. As we show in this study, XPA interacts with DNA in a dynamic manner going through episodic periods of pauses and then short range motion. Thus, one can easily envision initial binding to the damage site and then XPA's migration away from the damaged site facilitated by its interaction with other factors including XPB and XPD as part of TFIIH. This type of limited diffusion away from damage has been observed for Rad4 and PARP1 from our lab and most recently for XPC from the Scharer and Lee laboratories¹. We have made this point in the discussion (lines 387-392).

One point that may be of interest for such discussion is that XPA higher affinity for a “bent” DNA helix doesn't necessarily translate in a lesion recognition role. Indeed, within the current (and widely/generally accepted) view of the NER excision complex, XPA binds at one of the two junctions, either 3' or 5', not yet 100% proven which one, there are arguments/data supporting each. From a structural point of view looking at the two single strands opening in opposite directions, the junction is a bent DNA construct, not too different from a bent DNA double helix if the protein binds only one of the strands. Within this framework, it is possible that XPA has a higher affinity for bent DNA constructs, because it needs to locate at a junction and not because it is involved in the damage detection step. Maybe the authors should comment also on this point.

We have updated the discussion (lines 329-331) to acknowledge that a bent DNA helix is a feature of both DNA lesions (such as AAF) and NER intermediates. With regard to a 5' or 3' junction, the idea that XPA is situated at the 5' junction of the lesion site is supported by recent Cryo-EM data from the Cramer laboratory showing a strong extended alpha helix interaction with XPB in TFIIH. Also, previous work showing XPA's interaction with the 5' nuclease ERCC1/XPF further supports the notion that XPA is interacting with a fork structure.

In summary, the topics of discussion I suggested above represent a minor revision that may require a slight change in this specific direction/aim. Nevertheless, I believe that such additional discussion would add to an already outstanding work and valuable contribution to the field. For this reason, with these minor revisions addressed, I strongly support the publication of the manuscript in Nature Communications.

Minor point:

Reference from Kokic et al, bioRxiv page 3 line 75 should be listed with all other references for consistency.

This paper was published after our initial submission and we have updated the citation².

Reviewer #2

This manuscript reports on the use of single molecule studies of XPA binding to damaged and undamaged DNA. The authors phrase their study in the context of nucleotide excision repair (NER), where XPA is known to function. However, this is a purely biophysical study of the protein in isolation, so the results only remotely relate to the role of XPA in NER. The data confirm previously published work of high affinity binding to an AAF lesion, that XPA binds DNA substrates as a monomer, and that DNA is bent by $\sim 60^\circ$ at sites where XPA is bound. Many different modes of linear diffusion of XPA on damaged and undamaged DNA, and the authors place these results in a model of XPA functioning in the search for distorted duplex DNA associated with damage.

The manuscript is written in clear language with a good logical flow. The experimental design, data acquisition, and analyses are technically sound and appear to be of high caliber. However, there is a fundamental flaw in this study when it comes to transferring the observations of the effects of isolated XPA to the broader landscape of the NER pathway. It is well accepted (and even summarized in the Introduction) that XPA is not present at sites of DNA damage until the damage is recognized by other NER proteins and the TFIIH complex is recruited. TFIIH opens the damaged duplex and unwinds it around the site of damage, then XPA is recruited along with RPA. At this point, XPA is not freely diffusing in solution searching for damage in a duplex, hence it is completely unclear how the results pertain to XPA function in NER.

The results do show that the presence of DNA damage affects how isolated XPA binds and diffuses on the unencumbered substrate. But the absence of the many other components of the large NER machinery (over 30 different proteins) limits the scope and significance of their findings. Given these limitations, it is misleading to direct the Introduction to NER and to try and relate the data to NER, and then only in the last sentence state that the model presented is a 'starting point' and that XPA may well behave quite differently when bound to other proteins.

As we discuss in our response to Reviewer 1, we do not claim to have defined the role of XPA in NER, or to have shown exactly how XPA finds its target in a living cell. We have emphasized in the discussion that “we present our model as an important starting point, illustrating XPA-DNA binding behavior and diffusive properties in isolation.”

Overall, the results obtained are relevant to biophysical investigations of DNA repair, but only remotely to the role of XPA in NER. Potential functions of XPA outside of NER have been reported recently. Do one or more of these functional roles better relate to these data? As presented here, the significance and potential impact of these results make this manuscript better suited to a more specialized journal.

Concerns

1. The Introduction (pg 3, lines 57-62) states: "XPA has been reported to display specificity for UV-irradiated DNA 6-4PPs, N-(2'-deoxyguanosin-8-yl)-2-acetylaminofluorene (dG-C8-AAF), cisplatin, benzo[a]pyrene, non-hybridized bases, and other artificially distorted substrates. Additional studies revealed that XPA also binds preferentially to partially single stranded DNA and forked substrates more closely resembling unwound NER intermediates. It has been previously proposed that XPA binding to damaged DNA induces conformational changes in the protein that are associated with different binding modes." The experimental design and presentation of results in this manuscript are not consistent with the well accepted role of XPA as a scaffold organizing the many partner proteins in NER complexes through protein-protein interactions.

Please see comments to Reviewer #1 with regard to XPA's role in NER. We have softened both the introduction and discussion. Briefly, we would like to emphasize that while we do not disregard the evidence for XPA acting as a scaffold protein, this is not mutually exclusive with its ability to recognize NER lesions in dsDNA. As noted by the reviewer, we have intentionally studied purified protein and purified DNA substrates in order to obtain a basic mechanistic understanding of XPA's interaction with damaged DNA. Our work does not preclude a role of XPA as a scaffolding protein in NER. We are simply trying to understand its stoichiometry, and DNA binding properties and whether XPA's proven specificity for DNA lesions (see our table starting on pg. 13 of this rebuttal) can be observed with long DNA substrates.

2. The findings on the stoichiometry of XPA binding to DNA are not new; it is broadly accepted that XPA does not dimerize. There are many studies supporting this, including the structure of truncated Rad14 protein, in which two molecules are bound to the damaged DNA duplex, but there are no direct protein-protein contacts between them.

We appreciate the fact that the Rad14 structure³ contains no direct protein-protein contacts for the dimer. However, conclusions made in this study report that the protein does bind lesions as a dimer, and this paper has been cited as evidence for dimeric binding by XPA. Additionally, Yang et al⁴ are also frequently cited in the recent literature as evidence of an XPA dimer. The new TFIIH structure² does show XPA as a monomer in the context of the verification bubble, which supports our conclusion in the context of XPA **with other proteins** on DNA, but we respectively disagree that it is "broadly accepted that XPA does not dimerize." Below are some other papers that report XPA dimers on DNA:

Yang et al, 2006⁵

Liu et al, 2005⁶

Brown et al, 2010⁷

Gilljam et al, 2012⁸

3. The experimental design is lacking one important control: a DNA substrate with a static bend but no DNA damage (introduced for example by the uneven lengths of strands). This would inform the role of XPA in DNA damage recognition. On pg 11 (line 268) the heading says Presence of UV damage slows the rate of mode switching, leading to longer phases. It is difficult to assess if the differences observed are caused truly by the helix-distorting DNA damage or by the DNA distortion on its own attracting XPA to the substrate. While the authors discuss this possibility on pg 15, line 350, and it is consistent with lack of binding to a static bent DNA reported previously by Yang and colleagues (Yang et al., 2006), an appropriate control is required. This is especially important in the context of NER, as XPA may preferentially bind DNA that has already been bent by other components of NER complexes.

Several binding studies on these types of substrates have been published and are cited in our manuscript (including papers by Missura et al⁹ and Yang et al⁵). These and other papers are further detailed in the table at the end of this document (pg. 13). We believe these studies do provide compelling support for the idea that XPA binds preferentially to bent DNA. However, we argue that these artificial substrates do not more closely resemble NER intermediates than they resemble NER substrates and, as such, do not necessarily resolve the debate of XPA's role in damage recognition/verification. Instead, they support the idea that XPA is able to recognize bent/flexible DNA, in agreement with our present results, and this propensity to bend DNA likely plays a role in both the search for DNA damage, as well as affinity for NER intermediates. We therefore feel strongly that additional experiments on bent DNA substrates are not warranted.

4. Pg 5, line 119 – Of all the complexes observed, 33% were bound to the ends of the DNA substrate. This is quite a high rate of non-specific binding. Even 22% for the AAF538 substrate seems to be quite a high ratio of non-specific to specific binding. Why? Some comment on this is required in the text. The occurrence of cases with two molecules bound at both ends also observed also need to be reported.

Many proteins bind to the ends of DNA molecules (we have added citations) and we do not believe this is non-specific binding, but that XPA does, in fact, recognize specific types of DNA structures (like unpaired DNA ends) beyond damage site distortion. We have confirmed that we never see any cases in which a 538 bp DNA molecule is bound at both ends simultaneously. We have acknowledged this in the text (lines 114 and 312).

5. Pg 11, line 248 – Results show that most (60-70%) of the XPA protein is in a stationary mode on tightrope substrates. Is the remaining 30-40% fraction of motile protein sufficient to conclude that XPA actively screens for DNA-helix distortion? Does this ratio change when more than one damage site per DNA substrate is introduced? How do the authors explain that the motile fraction of the protein changes modes, including pausing, but the stationary fraction does not?

Due to the trade-off between observing and recording multiple binding events over the course of an experiment versus watching one particle for greater than 300 seconds, it is possible that some of these non-motile molecules would have diffused if we had observed for longer periods of time. It is also important to point out that the addition of increased salt helps to mobilize some of these non-motile molecules, so we do not believe these are necessarily dead end complexes. Furthermore, we have performed tightrope experiments on UV lambda DNA with an increased

UV dose (80 J/m²), resulting in increased lesion density (~1 lesion every 550 bp). The experiments show that XPA pauses more frequently on this substrate.

6. Is there any speculation on what the pausing positions on the tightropes are?

We speculate in the manuscript (lines 237-240) about possible spontaneous damage or bent DNA sequences in the non-damaged substrate that may cause XPA to pause. To address this question further, we have performed tightrope experiments on λ DNA with an increased UV dose (80 J/m²). We see a consistent trend of increased time spent in the paused mode and an increased number of pause sites when comparing XPA behavior on non-damaged λ DNA to its behavior on the λ DNA treated with either 20 J/m² or 80 J/m² UV, suggesting that these pause sites in the UV-irradiated DNA are indeed due to interrogation of UV-induced photoproducts, (Figure 5d-e).

7. Pg 16, lines 391-399 – The authors speculate without any specific support that the disordered N- and C-termini may fold in and interrogate the DNA for lesions via DNA bending. Considering that XPA is known to function as the key scaffold for NER pre-incision complexes and the many studies showing that it interacts with many other NER proteins, it is illogical to speculate that the disordered N- and C-termini of XPA are important for DNA binding only. This speculation should be supported by data or completely deleted from the text.

We greatly appreciate this suggestion by the reviewer, and show in Figure 5, panels d and e, that we have now created a truncation variant of XPA (M98 through T239) which removed residues 1-97 on the N-terminus and 240-273 on the C-terminus. In our EMSA studies, this variant displays identical non-specific DNA binding affinity as full length XPA and a 1.6 fold decreased specificity for AAF adducts (see new Supplementary Figure 1). Single molecule analysis revealed that this truncated XPA shows few pausing events on highly damaged λ DNA (80 J/m² UV-C), behaving like full length XPA protein on non-damaged DNA.

Additional points.

1. Pg 3, line 68 – A zinc finger is a specific structural motif that is characterized by the coordination of one or more zinc ions in order to stabilize the fold; the term should be reserved for such motifs. XPA possesses a Zn-binding motif, but not a Zinc finger and the wording should be adjusted accordingly.

While this is a relatively minor point with regard to the data presented in our study, we respectively disagree with the reviewer's distinction. XPA does in fact contain a C4 type zinc finger motif. We understand that C4 is not a standard/common zinc finger motif. UniProt classifies C4 zinc fingers as a family without sequence similarity beyond a zinc coordinated by 4 cysteines, but as a zinc finger nonetheless (https://www.uniprot.org/help/zn_fing). The term "zinc finger" has been used to describe this feature of XPA in several published papers, including the structure of Rad14 in complex with damaged DNA (Koch et al PNAS 2015). Thus, we did not change this nomenclature.

2. Pg 3, lines 74-75 – The Kovic et al. manuscript is now published in Nature Comm.

Citation has been updated.

3. Pg 4, lines 99-100 – How do the K_D values compare to previously published data?

In line 96, we compare our results with previous K_D 's reported by Sancar and colleagues. We have assembled a table, beginning on pg. 13 of this document, of papers showing that XPA has affinity for specific DNA substrates and give the K_D 's when reported in the paper. However, it is not possible to directly compare our values to others, because substrates, protein preparations, buffer conditions, etc. differ.

4. Pg 6, line 150 – The expected molecular mass of XPA monomer is 32.6 kDa but on pg 3, line 66 XPA is described as a protein of molecular mass of 31.4 kDa. While the origin of this inconsistency (XPA-His vs XPA) is mentioned at the Figure 2 label, it should be also clarified in the main text.

We appreciate the reviewer finding this discrepancy. The manuscript has been updated to clarify size discrepancies of purified XPA. We now explicitly state the construct used for all experiments and its molecular weight for all stoichiometry experiments (His-fIXPA is 32.6 kDa and His-fIXPA-StrepII is 33.9 kDa).

5. Pg 7, line 164 – Wording should be changed for Because XPA is a small protein (...). For example, "XPA is a relatively small protein for precise AFM measurement (...)"

We appreciate the reviewer's suggestion, but line 164 (line 155 in the updated version) is not related to the precision of the AFM measurement, which is actually discussed in a previous section (lines 106-110). Instead, we are explaining that because the protein is small, the ratio of DNA/protein in the complex will be large enough to necessitate a method to subtract the DNA volume.

6. Pg 23, line 601 – XPA is known to bind a zinc ion, and the presence of Zn has an influence of the structure. Hence, use of EDTA in the buffer may affect the protein's fold, stability, and solubility. Were controls performed?

EDTA was only present in the tightrope buffer, which due to the light intensity and Qdots, can cause oxidation (not for EMSA or AFM). EDTA is routinely used in molecular biology assays to bind heavy metals and to avoid oxidation reactions, particularly at Cys residues, and has been used routinely in the literature in buffers examining the biochemistry of XPA. Some examples include:

Koch et al, 2015³: 1 mM EDTA in EMSA buffer

Fischer et al, 2014¹⁰: 1 mM EDTA in XPA purification/storage buffer and in EMSA buffer

Saijo et al, 2011¹¹: 1 mM EDTA in XPA storage buffer

Neher et al, 2010¹²: 1 mM EDTA in XPA purification/storage buffer

Tsodikov et al, 2007¹³: 1 mM EDTA in EMSA buffer

Reviewer #3 (Remarks to the Author):

The manuscript by Beckwitt et al. presents single molecule atomic force microscopy (AFM) and fluorescence studies of the nucleotide excision repair (NER) factor Xeroderma pigmentosum group A (XPA). The authors aim at characterizing conformational and dynamic properties of XPA-DNA complexes during lesion search and at a target lesion to understand the mechanism of lesion recognition by XPA.

XPA is an essential protein and mutations in XPA severely affect human health. XPA functions in the NER pathway that repairs a plethora of structurally and chemically diverse DNA lesions, including acetylaminofluorene (AAF), which is used in the study, and in particular and famously UV induced damages in the DNA. **The exact role of XPA in the NER mechanism is still controversial.** In addition, the full length protein in complex with DNA has not been accessible to X-ray crystallography due to the intrinsic disorder in parts of the protein. The questions addressed by the authors are therefore highly relevant and interesting. In their studies, the authors investigate XPA in isolation (without other NER components present), while in vivo XPA functions in NER in complex with the transcription factor IIH (TFIIH) helicase machinery. However, understanding the conformational dynamics of DNA interactions by the isolated XPA protein adds to our better appreciation of the role of XPA in NER lesion processing.

The authors strive to resolve the question of the oligomeric state of XPA, the DNA bending introduced by XPA in non-damaged DNA and at an AAF lesion, and the mode(s) of translocation on DNA in the absence and presence of lesions. Regarding the oligomeric state of XPA, contradictory findings had been reported in the past. Beckwitt and colleagues clearly establish the monomeric form of XPA to be the dominant species both in solution and when bound to DNA, using AFM and multi angle light scattering. AFM characterization of DNA bending by XPA revealed comparable bending at non-damaged and lesion sites, and the bending angle was similar to that observed in crystal structures (of the shortened construct). The authors argue with energetic cost of DNA bending as a means for XPA to distinguish between lesion and non-damaged DNA sites based on their different flexibilities, as has been proposed before by others for XPA and for other DNA repair proteins. Regarding XPA lesion search dynamics, the authors report several different modes of translocation on DNA and the frequent interchange between these different modes. They interpret the modes as XPA testing for damage (pausing), XPA carrying out one-dimensional diffusion on the DNA in its lesion search (short range motion), and XPA in a fast scanning mode (mid range or long range motion). In a model they summarize their conclusions that XPA lesion search involves three different states: XPA searches for lesions in a fast scanning complex with diffusion coefficient dependent on ionic strength of the buffer (which they interpret as an indication for hopping); XPA probes for lesions in a short range scanning complex with lower diffusion constants, which may be in part caused by DNA bending by the protein to probe for lesions; and XPA forms a strongly bent stable recognition complex at the lesion.

Overall, I find the topic and results of this study interesting and highly relevant and I mostly agree with their conclusions. I do however, have a few points of criticism listed below. After minor changes, I hence recommend the manuscript for publication in Nature Communications.

Specific points:

1. AAF binding specificity determination by AFM:

The distribution of XPA positions on the DNA does not really resemble a > 600-fold preference for lesion binding (when compared, for example, to distributions in Yang et al. NAR 2005 as cited by the authors in this context). I agree with the authors that there is specificity for AAF by XPA, but I think the value is overestimated due to underestimation of non-specific background binding. Differently expressed: I think the Gaussian width in the fit to the distribution is too broad, artificially increasing the specific area that results in the high specificity value. A smaller binning interval might more clearly show whether the maximum at the lesion position is truly this broad or whether the Gaussian shaped peak is in fact much narrower, but would likely require more data points. In fact, the specificity that this distribution reflects likely fits nicely with that obtained by the authors from their EMSA analyses. However, based on the high flexibility at an AAF lesion, it actually seems surprising that the specificity of XPA for this lesion should not be larger than observed here, if energetic preference of enhanced flexibility sites determines the probability of occupancy. Furthermore, enhancement of TFIIH lesion recognition through XPA, as suggested previously (and also mentioned by the authors) is difficult to imagine with the low to moderate recognition specificity compared to – for instance, that shown for the XPD helicase of TFIIH (admittedly for different lesions). These points could be addressed.

We appreciate this reviewer raising this point, but we are constrained by the typical binning routine supported in the literature of using the number of bins equal to the square-root of the total number of observations¹⁴. The relatively broad distribution around the mean is typical for AFM studies of protein-DNA interactions as we and others have observed. We have recently reviewed¹⁵ and we have specifically observed this type of data for Rad4¹⁶, PARP1¹⁷, APE1 and DNA polymerase β (unpublished). Thus, we do not feel that XPA is an outlier. The reviewer raises a good point and we strongly believe that XPA interaction with other proteins can help to increase specificity through protein-protein interactions and additional modification or the structure of the DNA. However, this work is outside the scope of this present body of data.

2. XPA volume analyses by AFM:

While I appreciate the thoroughness of their volume calibration for their peak force AFM, I think this section could mostly be moved to the Supplement. It is obviously indispensable, but does not add to the manuscript story. The authors analyzed the volumes of two proteins, APE1 and Pol(beta) bound to DNA, to test the validity of their volume calibration also for DNA bound protein volumes. In this context, I also think that the originally measured volumes for APE1 and Pol(beta) (before DNA volume subtraction) ought to be given in the text, in addition to the final volumes after DNA subtraction. After the thorough consideration of the contribution from DNA binding mode by APE1 and Pol(beta) to overall measured volume, it would recommend itself to also discuss this for the protein of interest in this study, XPA to support the approach of subtracting the DNA volume to determine the molecular mass of the DNA bound complex. Again, I think the originally measured volumes (prior to DNA volume subtraction) should also be given, as well as the DNA volumes.

We appreciate this suggestion and we have moved some of these data to the supplemental data section. We agree with showing both the volume prior to and after subtraction, and have added this data to Figure 3 and Supplementary Figure 4.

3. DNA bend angle analyses by AFM:

In the bend angle distributions for DNA in the absence of protein (Figures 4C and D), the first interval seems to be only half the size of all other intervals since it is centered at zero degrees bending. If this is the case, then at least in the histogram shown in (C), correction to full interval size for the first range of bend angles (0 to 5 or 10 or whatever degrees) would likely result in a distribution with exponential decay from 0 degrees (rather than the 5.6 degrees maximum reported here). For the distribution in (D), at the AAF position, the first maximum very likely remains at the second bin, ie. larger than 0 degrees. In addition, for a better comparison between the two distributions, ideally the interval (bin) size should be the same, which would then be limited by the sample with the smallest number of datapoints (also for Figures 4 G and H). Also, for smaller bin sizes, the distributions in Fig. 4 G and H may well reveal more than one bend angle species, as also suggested by the example images (but smaller binning will be difficult due to small samples sizes). Two different conformational states in lesion search and recognition by XPA would seem interesting also in the context of the final model proposed by the authors. However, none of this will drastically change the story.

As suggested by this reviewer, we have replotted all DNA bend angles in Figure 4, and we appreciate the reviewer suggesting this change. The non-damaged DNA bend angle is centered on zero degrees.

4. DNA tightrope assay:

a. The authors state that the proportion of immobile XPA is not affected by the DNA substrate. However, Figure 5B shows ~90% versus ~70% immobile XPA complexes on DNA (including dissociating and persistent molecules) for AAF versus UV or non-damaged substrate. The statistical non-significance level seems surprising! The authors state that the density of UV lesions in their substrate can be expected to be comparable to that in the AAF substrate (approximately one lesion per ~2,000 bp), yet the percentage of immobile complexes appears to differ between the UV and AAF substrates. Can the type of lesion explain the different behavior?

With regard to the statistical question, while the proportions of stationary/motile particles appear different between UV and AAF tightropes, the χ^2 test, which is the appropriate statistic for analysis of all groups, shows no significant difference between groups. While the 6-4PP is more helix-distorting than the CPD, we have shown that Rad14 (XPA homolog in yeast) has higher affinity for an AAF than for either photoproduct³ and this may, in part, explain the behavior observed in the current study. It is possible that an AAF lesion can more readily induce a conformational change in XPA to promote pausing and stationary complexes. While we agree with the reviewer, we have erred on the side of caution when drawing conclusions from these comparisons. It is interesting to note in Figure 5d, where the motile fraction of XPA molecules is investigated in more detail, that lesion density alters the type of motion in a statistically significant way. Please see also point 5 to Reviewer 2.

b. The lack of medium/long range scanning species for AAF versus non-damaged or UV substrate is striking. Long/medium range motion is defined by the authors as movement over > 690 nm, the distance between two AAF lesions and approximate average distance between two UV lesions assuming the above stated density. Exclusively short range scanning by XPA is

hence consistent with XPA not proceeding over an AAF lesion. Again, could this depend on the type of lesion, since for the UV substrate a significant population of medium/long range scanning molecules was observed? In contrast to AAF, CPD lesions, as one type of UV lesion, do not strongly distort DNA.

The reviewer is correct to point out the lack of long-distance motion on the AAF arrays. In order to look at this in more detail, we have done an additional experiment with λ tightropes treated with 80 J/m² UV-C. While there is a slight decrease in long-range diffusion under these conditions, the most significant response to lesion density and type of lesion is time spent in the paused mode (Fig. 5d). With increasing UV, where the 6-4PPs are at higher density (approximately 1 6-4PP per 2.2 kbp for 80 J/m² treated tightropes), we see an increased amount of pausing, comparable to the AAF array (1 AAF per 2 kbp).

c. Many of the sub-species do not seem to contain enough data points for statistical analyses. For example $n=3$ for long range diffusion coefficients of pooled UV and non-damaged substrates, or $n=3$ for maximum displacement at 1 M salt concentration. I appreciate that data collection with this type of single molecule experiment is time consuming, but would be careful to draw conclusions from such small sample sizes.

We understand this limitation, and have tried to be cautious in drawing conclusions. Combining mid-range and long-range modes into one category (see point below) has improved the sample sizes in several cases.

d. I do not agree with the classification of long range scanning particles based on the example trace shown in Figure 5D. In this example, the “long range” trace seems to consist simply of two medium range movements interrupted by a clear pausing plateau. I would suggest to pool the data from medium range and long range modes. In the population distributions (e.g. Figure 6F) the long range species never seems to truly add any additional information. Also, the issue of not enough data points for some of the different species (point above) may automatically be removed when pooling medium and long range scanning particles.

We appreciate this comment and have now combined mid/long-range diffusion into one mode (called long-range).

e. I also wonder if the salt concentration effect on the medium/long range but not on the short range moving species really supports hopping as the mode of translocation in the medium/long range species and one dimensional diffusion for the short range species. Regarding the short range species, the distribution of short range diffusion coefficients (Figure 6E) appears to indicate one species consistent with one dimensional diffusion (below the limit line for diffusion) and a second with higher diffusion constants (above the limit line for diffusion). Regarding the medium/long range species, it seems to me that hopping is only one possible explanation of the effect observed at higher salt concentration. Could the salt concentration not instead simply affect stability of DNA interactions by the protein? These interactions seem to be majorly electrostatic when looking at published structural data. The example in Supplementary Figure 5c showing transition from paused to long range diffusion when increasing the salt concentration to 1M would be consistent with this.

The reviewer is correct in that increased ionic strength will affect electrostatic interactions. The reason that increasing ionic strength increases rates of diffusion during hopping is that, hops during micro-dissociation of XPA from the DNA get longer due to shielding of the electrostatic potential of the negatively charged phosphates. It is generally accepted in the single molecule field that if increasing ionic strength increases diffusion rates, then the protein is displaying hopping^{18,19}.

f. I wonder if DNA bending as seen in AFM applies for the tightrope assay or if the DNA is “fully” stretched between the anchoring beads? If XPA would not be able to bend the DNA as part of its lesion probing, this would affect the time spent probing for lesions (ie. bending the DNA, ie. in the paused state as well as in the short range linear diffusion state). The authors mention that other proteins previously tested in this assay did not show the changing between translocation modes seen for XPA. Were any of these proteins expected to bend the DNA?

We have clarified in the text that DNA is not fully stretched in the DNA tightropes, but only elongated to 90% of its contour length. We and others have used this flow cell set-up successfully to study other DNA-bending proteins, including Rad4-Rad23¹⁶.

Table 1. Published XPA-DNA Interactions demonstrating DNA damage and/or structure-specific specificity.

Unless otherwise noted: studies were performed using full-length wild-type human protein (recombinant, purified), fold specificities are reported as overall binding between substrates, and DNA substrates were prepared using short oligonucleotides (less than 60 bp or nt).

Paper	Substrate	K_D	Specificity	Method
Robins et al, 1991. ²⁰	UV-irradiated dsDNA (9 kJ/m ²) UV-irradiated dsDNA (0 - 9 kJ/m ²) Non-damaged ssDNA Non-damaged dsDNA		~1,000-fold specificity ^a for UV damage over non-damaged dsDNA. Fluence-dependent affinity for UV damage. Lower affinity for ssDNA than for dsDNA.	Filter binding ^{b,c}
Jones and Wood, 1993. ²¹	UV-irradiated dsDNA (6 kJ/m ²) UV-irradiated dsDNA (0-6 kJ/m ²), treated with CPD photolyase Cisplatin-treated dsDNA Psoralen-treated dsDNA Non-damaged dsDNA Non-damaged ssDNA, circular Non-damaged dsDNA, circular	333 nM 1.67 μ M	~300-fold specificity ^a for 6-4PP over dsDNA. ~4-fold specificity for circular ssDNA over circular dsDNA. Higher affinity for cisplatin than for non-damaged dsDNA. No specificity for CPD or psoralen adducts.	Electrophoretic mobility shift assay (EMSA) ^d
Asahina et al, 1994. ²²	UV-irradiated dsDNA (8 kJ/m ²) Cisplatin-treated dsDNA OsO ₄ -treated dsDNA Non-damaged ssDNA Non-damaged dsDNA		Specificity for UV-treated DNA over non-damaged. Specificity for dsDNA over ssDNA. Higher affinity for cisplatin than OsO ₄ -treated DNA; specificity for both over non-damaged.	Filter binding ^e
Li et al, 1995. ²³	UV-irradiated dsDNA (600 J/m ²) Non-damaged dsDNA		Specificity for UV-treated DNA over non-damaged.	Immobilized DNA template assay ^f

Paper	Substrate	K_D	Specificity	Method
Kuraoka et al, 1996. ²⁴	UV-irradiated dsDNA (8 kJ/m ²) Cisplatin-treated dsDNA Non-damaged dsDNA		Specificity for UV-treated and cisplatin-treated DNA over non-damaged.	Filter binding ⁹
Nocentini et al, 1997. ²⁵	UV-irradiated dsDNA (1 kJ/m ²) Non-damaged dsDNA		Specificity for UV-treated DNA over non-damaged.	Filter binding
Buschta-Hedayat et al, 1999. ²⁶	AAF-adducted dsDNA (-)-cis-B[a]P-adducted dsDNA (-)-trans-B[a]P-adducted dsDNA dsDNA with 3 nt MM dsDNA with 1 nt MM C4' pivaloyl-adducted dsDNA C4' pivaloyl-adducted dsDNA, adduct in 3 nt MM 3-nitroindole-modified dsDNA 5-nitroindole-modified dsDNA Non-damaged ssDNA Non-damaged dsDNA		Specificity for AAF and B[a]P over non-damaged dsDNA. Specificity for C4' pivaloyl adduct within bubble, but not in dsDNA. Higher affinity for 3 nt mismatch than 1 nt mismatch, specificity for both over dsDNA. Lower affinity for ssDNA than for dsDNA. Higher affinity for 5-nitroindoles than for 3-nitroindoles, specificity for both over non-modified dsDNA.	EMSA
Wakasugi et al, 1999. ²⁷	6-4PP-modified dsDNA Non-damaged dsDNA	6 nM 420 nM	~70-fold specificity ^a for UV-treated DNA over non-damaged.	EMSA

Paper	Substrate	K_D	Specificity	Method
Wang et al, 2000. ²⁸	6-4PP-modified dsDNA	21 nM	~4.5-fold specificity for ssDNA over dsDNA. ~3-fold specificity for 6-4PP. ~1.3-fold specificity for CPD.	Surface plasmon resonance
	CPD-modified dsDNA	46 nM		
	Non-damaged ssDNA	13 nM		
	Non-damaged dsDNA	58 nM		
Mustra et al, 2001. ²⁹	dsDNA with MMC interstrand XL		~2-3 fold specificity for MMS crosslink over non-damaged dsDNA.	EMSA
	Non-damaged dsDNA			
Hey et al, 2001. ³⁰	Cisplatin-adducted dsDNA, 3' FL	415 nM	~3-fold specificity for cisplatin, ssDNA loop, mismatched bubble, and ssDNA with mixed bases over non-damaged dsDNA. ~1.5-fold specificity for pyrimidine-rich ssDNA. No specificity/worse binding to purine-rich ssDNA compared to dsDNA.	Anisotropy
	dsDNA with 6 nt MM, 3' FL	380 nM		
	dsDNA with 3 nt insert on one strand, 3' FL	350 nM		
	Non-damaged ssDNA, mixed bases	355 nM		
	Non-damaged ssDNA, AG-rich	> 3 μ M		
	Non-damaged ssDNA, TC-rich	786 nM		
Non-damaged dsDNA, 3' FL	1.15 μ M			

Paper	Substrate	K_D	Specificity	Method
Missura et al, 2001. ⁹	Cisplatin-adducted dsDNA		Specificity for cisplatin, but not for the dinuclear analogue, over non-damaged DNA. Specificity for non-hybridized substrates as follows: dsDNA insert > ssDNA insert > mismatch bubble > dsDNA. Specificity for forked substrates as follows: 4-way dsDNA > 3-way dsDNA > Y > non-damaged dsDNA. Authors note "extraordinary affinity" of XPA for 4-way and 3-way dsDNA junctions. Higher affinity for 5-nitroindoles than for 3-nitropyrroles, specificity for both over non-modified dsDNA. No affinity for ssDNA.	EMSA
	Dinuclear cisplatin analogue-adducted dsDNA (Pt-Pt)			
	dsDNA with 3 nt MM			
	dsDNA with 3 nt insert on one strand			
	dsDNA with 3 bp insert on one strand			
	Y shaped DNA			
	3-way dsDNA junction			
	4-way dsDNA junction			
	3-nitropyrrole-modified dsDNA			
	5-nitroindole-modified dsDNA			
	Non-damaged ssDNA			
	Non-damaged dsDNA			
Iakoucheva et al, 2002. ³¹	dsDNA with 4 nt MM, 5' FL	158 nM	~5-fold lower affinity for mismatch compared to dsDNA.	Stop flow ^h
	Non-damaged dsDNA, 5' FL	28.9 nM		
	Non-damaged dsDNA, 5' FL	24.4 nM		
Reardon and Sancar, 2003. ³²	6-4PP-modified dsDNA	150 nM	~1.5-fold specificity for 6-4PP over non-damaged dsDNA. No specificity for CPD. Note: authors report similar fold specificity for RPA and XPC on same substrates.	EMSA
	CPD-modified dsDNA	210 nM		
	Non-damaged dsDNA	220 nM		

Paper	Substrate	K_D	Specificity	Method
Liu et al, 2005. ⁶	AAF-adducted dsDNA, 5' FL	714 nM (K_{D1}), 55 nM (K_{D2})	Note: authors report positive cooperativity (Hill = 1.9)	Anisotropy
	AAF-adducted dsDNA Non-damaged dsDNA	200 nM	Higher affinity for dG-C8-AAF than for non-damaged dsDNA.	EMSA
Brabec et al, 2006. ³³	Cisplatin-adducted dsDNA, 1,3-GTG		4-5-fold specificity for 1,2-GG adducts, when flanked T or A bases, over non-damaged dsDNA. ~2-fold specificity for 1,2-GG when flanked by C's. Less than 2-fold specificity for 1,3-GTG adducts.	EMSA
	Cisplatin-adducted dsDNA, 1,2-GG Non-damaged dsDNA			
Camenisch et al, 2006. ³⁴	Cisplatin-adducted dsDNA		Specificity for cisplatin over non-damaged dsDNA. Specificity for forked substrates as follows: 4-way dsDNA > 3-way dsDNA > Y > non-damaged dsDNA.	EMSA
	Y shaped DNA			
	3-way dsDNA junction			
	4-way dsDNA junction Non-damaged dsDNA			

Paper	Substrate	K_D	Specificity	Method
Yang et al, 2006. ⁵	Y-shaped DNA 5' overhang 3' overhang Y-shaped DNA dsDNA with 6 nt MM AF-adducted DNA, lesion in 6 nt MM AAF-adducted DNA, lesion in 6 nt MM Aminopyrene-adducted DNA, lesion in 6 nt MM 6-4PP-modified DNA, lesion in 6 nt MM dsDNA with 3, 4, 5, 8, 10, or 12 nt MM G[8,5-Me]T XLed dsDNA dsDNA with two, three, or four-carbon tether XL at GG Non-damaged ssDNA Non-damaged dsDNA	49 nM	Similar affinity for 3' overhang and 5' overhang. Similar affinity for all mismatch bubbles, with or without lesion. Higher affinity for bubbles with 8 or more mismatched bases. Specificity for F[8,5-Me]T crosslink over non-damaged DNA. No specificity for intrastrand crosslinks formed by carbon tethers. No affinity for non-damaged ssDNA or dsDNA.	Anisotropy EMSA
Mustra et al, 2007. ³⁵	dsDNA with MMC interstrand XL Non-damaged dsDNA		~2-fold specificity for MMC XL over non-damaged DNA.	EMSA
Krasikova et al, 2008. ³⁶	FL-dUMP-adducted dsDNA		Specificity for Flu-dUMP over non-damaged dsDNA	EMSA

Paper	Substrate	K_D	Specificity	Method
Brown et al, 2010. ⁷	AAF-adducted dsDNA	44 nM	Similar affinity for AAF and thymine glycol. Specificity for both over non-damaged dsDNA.	EMSA
	Thymine glycol-modified dsDNA	48 nM		
	Non-damaged dsDNA			
Sugitani et al, 2014. ³⁷	Y-shaped DNA, 5' FL (label on dsDNA end)	290 nM	~6-fold specificity for Y-shaped DNA over ssDNA and dsDNA.	Anisotropy
	Non-damaged ssDNA, 5' FL	1.5 μ M		
	Non-damaged dsDNA, 5' FL	1.7 μ M		
Koch et al, 2015. ³	AAF-adducted dsDNA		Higher affinity for dG-C8-AAF and FITC than for cisplatin.	EMSA
	FITC-adducted dsDNA			
	Cisplatin-adducted dsDNA			
	AAF-adducted dsDNA	135 nM	No specificity for CPD or 6-4PP.	EMSA ⁱ
	FITC-adducted dsDNA			
	Cisplatin-adducted dsDNA			
	6-4PP-modified dsDNA			
CPD-modified dsDNA				
Ebert et al, 2017. ³⁸	AF-adducted dsDNA		Higher affinity for dG-C8-AAF than for dG-N ² -AAN or dG-C8-AF. Higher affinity for all lesions compared to non-damaged dsDNA.	EMSA
	AAF-adducted dsDNA			
	AAN-adducted dsDNA			
	Non-damaged dsDNA			

NOTES

- ^a Specificity calculated to account for nonspecific bases in damaged substrate²¹
- ^b XPA fractionated from calf thymus
- ^c 779 bp and 2961 bp (mixed) DNA
- ^d Linear DNA substrates, 258 bp; circular substrates, M13 DNA
- ^e 7250 bp DNA
- ^f 622 bp and 485 bp (mixed) DNA
- ^g 2686 bp DNA
- ^h *Xenopus laevis* XPA
- ⁱ *Saccharomyces cerevisiae* Rad14 (10-end)

ABBREVIATIONS

6-4PP – (6-4) pyrimidine-pyrimidone photoproduct
AF – 2-aminofluorene
AAF – 2-acetylaminofluorene
AAN – N²-acetylnaphthyl
B[a]P – Benzo[a]pyrene
CPD – Cyclobutane pyrimidine dimer
dG-C8-AAF – N-(2'-deoxyguanosin-8-yl)-2-acetylaminofluorene
dsDNA – Double-stranded DNA
FITC – Fluorescein isothiocyanate
FL – Fluorescein
 K_D – Equilibrium dissociation constant
MM – DNA mismatch
MMC – Mitomycin C
XL – Crosslink

REFERENCES

- 1 Cheon, N. Y., Kim, H. S., Yeo, J. E., Scharer, O. D. & Lee, J. Y. Single-molecule visualization reveals the damage search mechanism for the human NER protein XPC-RAD23B. *Nucleic Acids Res* **47**, 8337-8347, doi:10.1093/nar/gkz629 (2019).
- 2 Kokic, G. *et al.* Structural basis of TFIIH activation for nucleotide excision repair. *Nat Commun* **10**, 2885, doi:10.1038/s41467-019-10745-5 (2019).
- 3 Koch, S. C. *et al.* Structural insights into the recognition of cisplatin and AAF-dG lesion by Rad14 (XPA). *Proceedings of the National Academy of Sciences of the United States of America* **112**, 8272-8277, doi:10.1073/pnas.1508509112 (2015).
- 4 Yang, Z. G., Liu, Y., Mao, L. Y., Zhang, J. T. & Zou, Y. Dimerization of human XPA and formation of XPA2-RPA protein complex. *Biochemistry* **41**, 13012-13020 (2002).
- 5 Yang, Z. *et al.* Specific and efficient binding of xeroderma pigmentosum complementation group A to double-strand/single-strand DNA junctions with 3'- and/or 5'-ssDNA branches. *Biochemistry* **45**, 15921-15930, doi:10.1021/bi061626q (2006).
- 6 Liu, Y. *et al.* Cooperative interaction of human XPA stabilizes and enhances specific binding of XPA to DNA damage. *Biochemistry* **44**, 7361-7368, doi:10.1021/bi047598y (2005).
- 7 Brown, K. L. *et al.* Binding of the human nucleotide excision repair proteins XPA and XPC/HR23B to the 5R-thymine glycol lesion and structure of the cis-(5R,6S) thymine glycol epimer in the 5'-GTgG-3' sequence: destabilization of two base pairs at the lesion site. *Nucleic Acids Res* **38**, 428-440, doi:10.1093/nar/gkp844 (2010).
- 8 Gilljam, K. M., Muller, R., Liabakk, N. B. & Otterlei, M. Nucleotide excision repair is associated with the replisome and its efficiency depends on a direct interaction between XPA and PCNA. *PLoS One* **7**, e49199, doi:10.1371/journal.pone.0049199 (2012).
- 9 Missura, M. *et al.* Double-check probing of DNA bending and unwinding by XPA-RPA: an architectural function in DNA repair. *EMBO J* **20**, 3554-3564, doi:10.1093/emboj/20.13.3554 (2001).
- 10 Fischer, J. M. *et al.* Poly(ADP-ribose)-mediated interplay of XPA and PARP1 leads to reciprocal regulation of protein function. *FEBS J* **281**, 3625-3641, doi:10.1111/febs.12885 (2014).
- 11 Saijo, M., Takedachi, A. & Tanaka, K. Nucleotide excision repair by mutant xeroderma pigmentosum group A (XPA) proteins with deficiency in interaction with RPA. *J Biol Chem* **286**, 5476-5483, doi:10.1074/jbc.M110.172916 (2011).
- 12 Neher, T. M., Shuck, S. C., Liu, J. Y., Zhang, J. T. & Turchi, J. J. Identification of novel small molecule inhibitors of the XPA protein using in silico based screening. *ACS Chem Biol* **5**, 953-965, doi:10.1021/cb1000444 (2010).
- 13 Tsodikov, O. V. *et al.* Structural basis for the recruitment of ERCC1-XPF to nucleotide excision repair complexes by XPA. *EMBO J* **26**, 4768-4776, doi:10.1038/sj.emboj.7601894 (2007).
- 14 Rivetti, C., Guthold, M. & Bustamante, C. Wrapping of DNA around the E.coli RNA polymerase open promoter complex. *EMBO J* **18**, 4464-4475, doi:10.1093/emboj/18.16.4464 (1999).
- 15 Beckwitt, E. C., Kong, M. & Van Houten, B. Studying protein-DNA interactions using atomic force microscopy. *Semin Cell Dev Biol* **73**, 220-230, doi:10.1016/j.semcdb.2017.06.028 (2018).
- 16 Kong, M. *et al.* Single-Molecule Imaging Reveals that Rad4 Employs a Dynamic DNA Damage Recognition Process. *Mol Cell* **64**, 376-387, doi:10.1016/j.molcel.2016.09.005 (2016).
- 17 Liu, L. *et al.* PARP1 changes from three-dimensional DNA damage searching to one-dimensional diffusion after auto-PARylation or in the presence of APE1. *Nucleic Acids Res* **45**, 12834-12847, doi:10.1093/nar/gkx1047 (2017).
- 18 Tafvizi, A., Huang, F., Fersht, A. R., Mirny, L. A. & van Oijen, A. M. A single-molecule characterization of p53 search on DNA. *Proceedings of the National Academy of Sciences of the United States of America* **108**, 563-568, doi:10.1073/pnas.1016020107 (2011).
- 19 Berg, O. G., Winter, R. B. & von Hippel, P. H. Diffusion-driven mechanisms of protein translocation on nucleic acids. 1. Models and theory. *Biochemistry* **20**, 6929-6948 (1981).
- 20 Robins, P., Jones, C. J., Biggerstaff, M., Lindahl, T. & Wood, R. D. Complementation of DNA repair in xeroderma pigmentosum group A cell extracts by a protein with affinity for damaged DNA. *EMBO J* **10**, 3913-3921 (1991).

- 21 Jones, C. J. & Wood, R. D. Preferential binding of the xeroderma pigmentosum group A
complementing protein to damaged DNA. *Biochemistry* **32**, 12096-12104 (1993).
- 22 Asahina, H. *et al.* The XPA protein is a zinc metalloprotein with an ability to recognize various
kinds of DNA damage. *Mutat Res* **315**, 229-237 (1994).
- 23 Li, L., Lu, X., Peterson, C. A. & Legerski, R. J. An interaction between the DNA repair factor XPA
and replication protein A appears essential for nucleotide excision repair. *Mol Cell Biol* **15**, 5396-
5402 (1995).
- 24 Kuraoka, I. *et al.* Identification of a damaged-DNA binding domain of the XPA protein. *Mutat Res*
362, 87-95 (1996).
- 25 Nocentini, S., Coin, F., Saijo, M., Tanaka, K. & Egly, J. M. DNA damage recognition by XPA
protein promotes efficient recruitment of transcription factor II H. *J Biol Chem* **272**, 22991-22994
(1997).
- 26 Buschta-Hedayat, N., Buterin, T., Hess, M. T., Missura, M. & Naegeli, H. Recognition of
nonhybridizing base pairs during nucleotide excision repair of DNA. *Proceedings of the National
Academy of Sciences of the United States of America* **96**, 6090-6095 (1999).
- 27 Wakasugi, M. & Sancar, A. Order of assembly of human DNA repair excision nuclease. *J Biol
Chem* **274**, 18759-18768 (1999).
- 28 Wang, M., Mahrenholz, A. & Lee, S. H. RPA stabilizes the XPA-damaged DNA complex through
protein-protein interaction. *Biochemistry* **39**, 6433-6439 (2000).
- 29 Mustra, D. J., Warren, A. J. & Hamilton, J. W. Preferential binding of human full-length XPA and
the minimal DNA binding domain (XPA-MF122) with the mitomycin C-DNA interstrand cross-link.
Biochemistry **40**, 7158-7164 (2001).
- 30 Hey, T., Lipps, G. & Krauss, G. Binding of XPA and RPA to damaged DNA investigated by
fluorescence anisotropy. *Biochemistry* **40**, 2901-2910 (2001).
- 31 Iakoucheva, L. M., Walker, R. K., van Houten, B. & Ackerman, E. J. Equilibrium and stop-flow
kinetic studies of fluorescently labeled DNA substrates with DNA repair proteins XPA and
replication protein A. *Biochemistry* **41**, 131-143 (2002).
- 32 Reardon, J. T. & Sancar, A. Recognition and repair of the cyclobutane thymine dimer, a major
cause of skin cancers, by the human excision nuclease. *Genes Dev* **17**, 2539-2551,
doi:10.1101/gad.1131003 (2003).
- 33 Brabec, V., Stehlikova, K., Malina, J., Vojtiskova, M. & Kasparkova, J. Thermodynamic properties
of damaged DNA and its recognition by xeroderma pigmentosum group A protein and replication
protein A. *Arch Biochem Biophys* **446**, 1-10, doi:10.1016/j.abb.2005.12.003 (2006).
- 34 Camenisch, U., Dip, R., Schumacher, S. B., Schuler, B. & Naegeli, H. Recognition of helical kinks
by xeroderma pigmentosum group A protein triggers DNA excision repair. *Nat Struct Mol Biol* **13**,
278-284, doi:10.1038/nsmb1061 (2006).
- 35 Mustra, D. J., Warren, A. J., Wilcox, D. E. & Hamilton, J. W. Preferential binding of human XPA to
the mitomycin C-DNA interstrand crosslink and modulation by arsenic and cadmium. *Chem Biol
Interact* **168**, 159-168, doi:10.1016/j.cbi.2007.04.004 (2007).
- 36 Krasikova, Y. S. *et al.* Interaction of nucleotide excision repair factors XPC-HR23B, XPA, and
RPA with damaged DNA. *Biochemistry (Mosc)* **73**, 886-896 (2008).
- 37 Sugitani, N., Shell, S. M., Soss, S. E. & Chazin, W. J. Redefining the DNA-binding domain of
human XPA. *J Am Chem Soc* **136**, 10830-10833, doi:10.1021/ja503020f (2014).
- 38 Ebert, C., Simon, N., Schneider, S. & Carell, T. Structural Insights into the Recognition of N(2) -
Aryl- and C8-Aryl DNA Lesions by the Repair Protein XPA/Rad14. *Chembiochem* **18**, 1379-1382,
doi:10.1002/cbic.201700169 (2017).

Reviewers' comments:

Reviewer #1 (Remarks to the Author):

The points I raised in my review of the first version of the manuscript have been exhaustively addressed by the authors. I am happy to recommend the current version of the manuscript for publication in Nature Communication as it is.

Elisa Fadda

Reviewer #2 (Remarks to the Author):

The revised version of this manuscript contains new data and additional analyses incorporated into the results, and modifications to the Introduction and Discussion sections. As noted previously, the experimental design, data acquisition, and analyses are technically sound and appear to be of high caliber, and these points hold for the new results. Because XPA functions as part of large multi-protein machinery, the limitations of studying the protein in isolation as noted previously have not been addressed.

The authors made changes to the text in response to various criticisms. They utilize a comment made by reviewer #3 to justify their reticence to acknowledge the prevailing view that XPA is not present at sites of DNA damage until the damage is recognized by other NER proteins and the TFIIH complex is recruited to open the DNA duplex. It is perfectly acceptable to state, as Reviewer 3 noted, that the role of XPA in NER remains controversial. Hence, it is also acceptable for the authors to believe that XPA plays a role in recognizing the presence of damage. Information about different potential roles for XPA is now included, but the manuscript needs to state more clearly that the role of XPA is controversial, in the Abstract, on page 2, line 47, and in the Discussion. Because there is controversy the term damage search in the title is inappropriate because this assumes one side of the controversy.

Similarly, in interpreting their data in the Discussion section, the authors need to acknowledge that the prevailing belief in the field is that TFIIH opens the damaged duplex and unwinds it around the site of damage, then XPA is recruited along with RPA, which means XPA is not freely diffusing in solution searching for damage in a duplex. Stating this, and then providing the alternate view that XPA may participate in the search for regions of the DNA containing lesions, strengthens their argument, by first providing a balanced perspective for discussing the relevance of the results, which would then set the stage for presenting the model in Figure 7. As noted previously, the authors could also add further support for the relevance of this study by noting that there is increasing evidence of XPA functions outside of NER for which the DNA binding properties characterized here could well be critical.

Another very critical concern is that there are a number of phrases in the manuscript that are imprecise and must be clarified if the manuscript is to be deemed suitable for publication.

1. It is important to state at the outset and incorporate terminology throughout the text that clarifies XPA recognizes aberration in the DNA duplex and not the lesion itself. Adding the phrase 'DNA containing' (e.g. 'XPA recognizes DNA containing damage') is a simple way to address this issue. This is central to how NER can repair such a diverse range of lesions.

2. The issue of 'specificity' in binding DNA also needs greater clarification. A simple way to remedy this problem would be to include the phrase 'relative to undamaged DNA'. The table provided in the response letter nicely shows how difficult it is to draw conclusions from data in the literature. For example, Ref 16 reports that XPA ss-ds DNA junctions with much higher affinity than DNA

containing lesions and the Kd value reported in that study is two-fold lower than the Kd value reported here for AAF containing DNA. But the methods used are very different. Addressing the specificity for damaged versus ss-ds junctions would be highly informative, especially in the context of defining the role of XPA in NER. The authors focused only on the binding of undamaged versus damaged DNA, so this should be made evident in presenting data and discussing the results.

3. Specific edits needed in the text (phrases to add are in all caps):

Page 2

Line 22- ...stoichiometry and THE ROLE IT MAY PLAY IN damage recognition ARE CONTROVERSIAL.

Lines 29-30- the last sentence of the Abstract drifts into speculation; it should be replaced by a summary of the main discoveries in this study.

Line 32- The word 'specifically' should be deleted because it causes the sentence to imply the NER machinery is recognizing the identity of the lesion.

Line 42- ... recognize THE PRESENCE OF A lesion ...

Line 49- ... RELATIVE TO UNDAMAGED DNA, XPA displays

Page 3

Line 53- replace 'report' with suggested and 'Additional' with Subsequent.

Line 57- replace 'be highly specific to' with accommodate.

Line 60- DNA CONTAINING lesions.

Line 70- The statement is not fully accurate: in Mer et al., Cell 2000, NMR studies of the XPA N-terminal domain (residues 1-98) showed it is disordered, and that its RPA binding motif folds into a helix when bound to RPA32C.

Line 74- Ref. 15 also reported binding of XPA to DNA as a monomer.

Page 4

Line 87- ... to DNA CONTAINING a ... and ... recognizes DNA CONTAINING AAF ..

Lines 91, 92 (and elsewhere in the manuscript)- Kd values from EMSA are not accurate to four significant figures.

Page 5

Line 127- these data do not provide any evidence of where XPA binds, so delete the phrase.

Line 152- include reference 15 for supporting binding of DNA as a monomer.

Page 8

Line 238- consider also contributions from transient bp opening

Line 257- given the differences in the data, the word "sharply" should be deleted.

Page 9

Line 266-267 This sentence is highly speculative with no evidence from this study to support it; it should be deleted.

Page 10

Line 307- ... substrates⁹, BUT OTHER STUDIES REPORTED BINDING TO DNA AS A MONOMER (15,21,28,36). Volumes ...

Line 320- ... formation, CONSISTENT WITH STUDIES REPORTING XPA BINDS TO DNA AS A MONOMER (15, 21, 28, 36).

Line 325- ... DNA helix(13,25) RELATIVE TO UNDAMAGED DNA, and ...

Line 328- There are no data supporting protein folding, so 'fold into' should be replaced by form.

Page 11

Line 354- the inclusion of the term "initial damage recognition" assumes XPA is involved in this step, but as noted above, this issue is controversial not proven. Since the authors provide no direct evidence that XPA is involved in initial damage recognition, this sentence needs to be rephrased.

Page 12

Line 381- ... specificity for DNA CONTAINING an AAF adduct ... and ... for DNA CONTAINING a CPD lesion ...

Line 383- replace "motion" with search.

Reviewer #3 (Remarks to the Author):

The revised manuscript by Beckwitt et al. has greatly improved and all my concerns have been addressed. I have only three minor comments.

- 1) The numbering of the Supplementary Figures 5-7 does not match their occurrence in the text.
- 2) In Supplementary Figure 7, it is not clear if His-flXPA-StrepII represents quantum dot labeled protein.
- 3) In Figure 4, I was surprised that the authors suggest that the $10.5^\circ \pm 7^\circ$ bend angle state in (d) for AAF-DNA is comparable to the 0° state in (c) for non-damaged DNA. These data look like they would easily pass any significance test. This is surprising since these are DNA substrates of identical sequence (apart from the presence or absence of the AAF lesion). The authors also specifically state in the text that they did measure DNA bending also at the 30% positions for the non-damaged DNA substrate (rather than at random positions along the DNA), which eliminates the possibility of sequence effects. Assuming that an experimenter bias towards larger bending at AAF-lesions can be excluded, I think these significantly different DNA bend angles are difficult to explain, but may not want to be completely ignored. Again, as in my previous comments, this does not change the story of the manuscript, which rather focuses on the larger bend angle of approximately 35° at the AAF lesion in the absence of XPA (which is clearly not present in the non-damaged DNA) compared to 60° in the XPA-DNA complex.

We are grateful for the continued efforts of each reviewer. New changes to the manuscript are described below and marked in blue text in the revised document.

Response to each reviewers' comments:

Reviewer #1 (Remarks to the Author):

The points I raised in my review of the first version of the manuscript have been exhaustively addressed by the authors. I am happy to recommend the current version of the manuscript for publication in *Nature Communication* as it is.

Elisa Fadda

We greatly appreciate that this reviewer is satisfied with the additional experiments and revisions to the manuscript, and that she recommends it for publication in *Nature Communications*. The manuscript and study have been strengthened by the helpful reviews.

Reviewer #2 (Remarks to the Author):

We greatly appreciate the precision that the new wording, suggested by the reviewer, brings to the revised manuscript.

The revised version of this manuscript contains new data and additional analyses incorporated into the results, and modifications to the Introduction and Discussion sections. As noted previously, the experimental design, data acquisition, and analyses are technically sound and appear to be of high caliber, and these points hold for the new results. Because XPA functions as part of large multi-protein machinery, the limitations of studying the protein in isolation as noted previously have are not addressed.

The authors made changes to the text in response to various criticisms. They utilize a comment made by reviewer #3 to justify their reticence to acknowledge the prevailing view that XPA is not present at sites of DNA damage until the damage is recognized by other NER proteins and the TFIIH complex is recruited to open the DNA duplex. It is perfectly acceptable to state, as Reviewer 3 noted, that the role of XPA in NER remains controversial. Hence, it is also acceptable for the authors to believe that XPA plays a role in recognizing the presence of damage. Information about different potential roles for XPA is now included, but the manuscript needs to state more clearly that the role of XPA is controversial, in the Abstract, on page 2, line 47, and in the Discussion. Because there is controversy the term damage search in the title is inappropriate because this assumes one side of the controversy.

We have gladly added an explicit statement to the manuscript (lines 22 and 50) acknowledging the controversy in the field regarding the role of XPA in NER. We believe that our current study demonstrates that XPA does participate in damage search under our conditions: XPA (1) binds with higher probability at AAF adducts by AFM, and (2) undergoes linear diffusion on DNA tightropes and exhibits increased pausing in response to UV damage. We have therefore left the title unchanged, but have been very cautious not to make any unsubstantiated claims regarding XPA behavior in the cell.

Similarly, in interpreting their data in the Discussion section, the authors need to acknowledge

that the prevailing belief in the field is that TFIIH opens the damaged duplex and unwinds it around the site of damage, then XPA is recruited along with RPA, which means XPA is not freely diffusing in solution searching for damage in a duplex. Stating this, and then providing the alternate view that XPA may participate in the search for regions of the DNA containing lesions, strengthens their argument, by first providing a balanced perspective for discussing the relevance of the results, which would then set the stage for presenting the model in Figure 7. As noted previously, the authors could also add further support for the relevance of this study by noting that there is increasing evidence of XPA functions outside of NER for which the DNA binding properties characterized here could well be critical.

We appreciate this concern and have added this point to the discussion (lines 342-345). We have intentionally presented Figure 7 as a working model of XPA behavior on DNA, based on what we have learned in this study. We have purposely not placed our model in the context of the greater NER pathway. The main purpose of the figure is to show (1) the different modes that XPA exhibits on DNA and (2) the possible role of the disordered domains in its motion.

Another very critical concern is that there are a number of phrases in the manuscript that are imprecise and must be clarified if the manuscript is to be deemed suitable for publication.

1. It is important to state at the outset and incorporate terminology throughout the text that clarifies XPA recognizes aberration in the DNA duplex and not the lesion itself. Adding the phrase 'DNA containing' (e.g. 'XPA recognizes DNA containing damage') is a simple way to address this issue. This is central to how NER can repair such a diverse range of lesions.

Additional clarification has been added to the manuscript in response to the specific issues listed below (point 3). We wish to clarify our terms here, as we agree with the reviewer at a fundamental level, but respectfully disagree on the semantics. With regard to this semantic difference between seeing the actual modified base or the altered structure induced by the modified base, we believe it is unfair to distinguish a helical distortion caused by a lesion from the lesion itself. For example, XPA recognition of helical aberrations may be what allows the protein to recognize and bind at an AAF adduct, as we have shown by AFM. We believe XPC and even UV-DDB probably work in a similar manner.

2. The issue of 'specificity' in binding DNA also needs greater clarification. A simple way to remedy this problem would be to include the phrase 'relative to undamaged DNA'. The table provided in the response letter nicely shows how difficult it is to draw conclusions from data in the literature. For example, Ref 16 reports that XPA ss-ds DNA junctions with much higher affinity than DNA containing lesions and the K_d value reported in that study is two-fold lower than the K_d value reported here for AAF containing DNA. But the methods used are very different. Addressing the specificity for damaged versus ss-ds junctions would be highly informative, especially in the context of defining the role of XPA in NER. The authors focused only on the binding of undamaged versus damaged DNA, so this should be made evident in presenting data and discussing the results.

We agree with this point and have made the suggested changes. While we are limited by space, we do state that XPA has specificity for ss-ds DNA junction in the introduction (lines 53-54). As the reviewer mentions, due to variation between methods and experimental conditions between papers, it is very difficult to draw conclusions and determine fold-differences by comparing separate studies. Ref 16 (Yang et al, 2006¹) reports that XPA binds to a forked Y-shaped DNA substrate with higher affinity than to dsDNA with a G[8,5-Me]T intrastrand crosslink.

Unfortunately, it is difficult to compare this to our results using AAF because the methods/conditions are different and because G[8,6Me]T does not destabilize the helix in the same way that an AAF adduct does. Yang et al did not do a direct comparison of AAF in dsDNA and a ss-dsDNA junction.

3. Specific edits needed in the text (phrases to add are in all caps):

Page 2

Line 22 - ...stoichiometry and THE ROLE IT MAY PLAY IN damage recognition ARE CONTROVERSIAL.

This change has been made.

Lines 29-30 - the last sentence of the Abstract drifts into speculation; it should be replaced by a summary of the main discoveries in this study.

We have softened the last sentence of the abstract to eliminate the speculation. Instead, we describe the working model only in terms of our current data (lines 29-30).

Line 32 (lines 32-33 in revised document) - The word 'specifically' should be deleted because it causes the sentence to imply the NER machinery is recognizing the identity of the lesion.

The sentence has been modified to reduce the possibility of mis-interpretation.

Line 42 (43) - ... recognize THE PRESENCE OF A lesion ...

This change has been made.

Line 49 (50-51) - ... RELATIVE TO UNDAMAGED DNA, XPA displays

This change has been made.

Page 3

Line 53 (54) - replace 'report' with suggested and 'Additional' with Subsequent.

We have changed "report" to "suggested." However, the term "subsequent" implies a timeline in which all recent data place XPA later in the NER pathway, but the chronology is not so straightforward. Some papers in support of an early role for XPA in NER: Wood, 1999²; Sancar et al, 2004³; Liu et al, 2005⁴; Koch et al, 2015⁵. Some papers in support of a later role for XPA in NER: Sugasawa et al, 1998⁶; Volker et al, 2001⁷; Rademakers et al, 2003⁸; Sugitani et al, 2016⁹.

Line 57 (58) - replace 'be highly specific to' with accommodate.

This change has been made.

Line 60 (61) - DNA CONTAINING lesions.

We have changed "stalling at a lesion" to "stalling on damaged DNA."

Line 70 (72-73) - The statement is not fully accurate: in Mer et al., Cell 2000, NMR studies of the XPA N-terminal domain (residues 1-98) showed it is disordered, and that its RPA binding motif folds into a helix when bound to RPA32C.

We appreciate this point and have updated this sentence (72-73 in revised manuscript) to clarify that we are only referring to resolved structures in the RCSB PDB.

Line 74 (76) - Ref. 15 also reported binding of XPA to DNA as a monomer.

We have added Ref 15 (Sugitani et al, 2017¹⁰) to this point, as well as a little more discussion on the controversy of XPA-DNA stoichiometry (lines 76-77).

Page 4

Line 87 (90-91) - ... to DNA CONTAINING a ... and ... recognizes DNA CONTAINING AAF .. We appreciate the reviewer's caution in claiming protein binding specificity and position. We have also been careful not to over-interpret our EMSA data and clarify several limitations of this method (lines 102-106). We have added the suggested edit on lines 90-91 ("XPA recognizes AAF-adducted DNA by EMSA"). However, the heading of this section refers to EMSA experiments as well as AFM experiments. AFM is a well-established tool for determining protein binding position along DNA molecules and the pioneering work by Erie and colleagues have demonstrated its use as a tool for studying specificity¹¹. We have recently reviewed this topic¹² and have listed some additional examples below:

Margeat et al, 1998¹³

Schulz et al, 1998¹⁴

Chen et al, 2002¹⁵

Buechner et al, 2014¹⁶

Josephs et al, 2015¹⁷

Sukhanova et al, 2016¹⁸

Please also see response to Line 127, below.

Lines 91, 92 (and elsewhere in the manuscript) (95, 247-248, 249, 251) - K_d values from EMSA are not accurate to four significant figures.

We appreciate this point and have updated all K_D values to report three significant figures.

Page 5

Line 127 (129-130) - these data do not provide any evidence of where XPA binds, so delete the phrase.

One of the major advantages of AFM imaging of protein-DNA interactions is that binding position *can be measured directly*. The purpose of Figure 1 is to provide evidence of where XPA binds, and thus we feel the phrase is justified as written. Please also see response to Line 87, above.

Line 152 (155) - include reference 15 for supporting binding of DNA as a monomer.

This change has been made.

Page 8

Line 238 (242) - consider also contributions from transient bp opening

We appreciate the thoughtful suggestion of transient bp opening, and have added this point to line 242.

Line 257 (260) - given the differences in the data, the word "sharply" should be deleted.

This change has been made.

Page 9

Line 266-267 (269) - This sentence is highly speculative with no evidence from this study to support it; it should be deleted.

This change has been made.

Page 10

Line 307 (311) - ... substrates⁹, BUT OTHER STUDIES REPORTED BINDING TO DNA AS A MONOMER (15,21,28,36). Volumes ...

We have added references to line 311.

Line 320 (322) - ... formation, CONSISTENT WITH STUDIES REPORTING XPA BINDS TO DNA AS A MONOMER (15, 21, 28, 36).

We feel that the addition of these citations in this line would be redundant after making the previously suggested references. Furthermore, this sentence is referring to the stoichiometry of free XPA, which is not the focus of the above-mentioned papers.

Line 325 (329) - ... DNA helix(13,25) RELATIVE TO UNDAMAGED DNA, and ...

This change has been made.

Line 328 (332) - There are no data supporting protein folding, so 'fold into' should be replaced by form.

This change has been made.

Page 11

Line 354 (361) - the inclusion of the term "initial damage recognition" assumes XPA is involved in this step, but as noted above, this issue is controversial not proven. Since the authors provide no direct evidence that XPA is involved in initial damage recognition, this sentence needs to be rephrased.

This is a working model based on the new data presented here. No other group has reported how XPA migrates on DNA and we feel strongly that this short range motion is a common theme in several DNA repair proteins. We thus find the comparison between the current data and previous reports to be relevant in discussing how our work may apply to NER. In this sentence (line 359 in the updated manuscript), we are purposely being vague about what protein is providing the initial damage recognition, and have edited the sentence to clarify this ambiguity. We clearly state XPA's scaffolding function and the initial damage recognition could be achieved by UV-DDB, XPC or even TFIIH – we do not know for AAF lesions.

Page 12

Line 381 (388-389) - ... specificity for DNA CONTAINING an AAF adduct ... and ... for DNA CONTAINING a CPD lesion ...

This change has been made.

Line 383 (391) - replace "motion" with search.

This change has been made.

Reviewer #3 (Remarks to the Author):

The revised manuscript by Beckwitt et al. has greatly improved and all my concerns have been addressed. I have only three minor comments.

- 1) The numbering of the Supplementary Figures 5-7 does not match their occurrence in the text. We have updated the numbering to reflect the order of appearance. Supplementary Fig. 7 → 5 (comparison of protein preps and Qdot labeling for AFM/tightrope experiments)

Supplementary Fig. 5 → 6 (comparison of phase lengths and diffusion coefficients)
Supplementary Fig. 6 → 7 (salt experiments)

2) In Supplementary Figure 7, it is not clear if His-flXPA-StrepII represents quantum dot labeled protein.

The legend of Supplementary Figure 5 (previously Supplementary Figure 7) has been updated to clarify that panels a-d are results of AFM experiments, and thus do not represent quantum dot-labeled protein. Panel e compares quantum dot labeling strategies for tightrope experiments.

3) In Figure 4, I was surprised that the authors suggest that the $10.5^\circ \pm 7^\circ$ bend angle state in (d) for AAF-DNA is comparable to the 0° state in (c) for non-damaged DNA. These data look like they would easily pass any significance test. This is surprising since these are DNA substrates of identical sequence (apart from the presence or absence of the AAF lesion). The authors also specifically state in the text that they did measure DNA bending also at the 30% positions for the non-damaged DNA substrate (rather than at random positions along the DNA), which eliminates the possibility of sequence effects. Assuming that an experimenter bias towards larger bending at AAF-lesions can be excluded, I think these significantly different DNA bend angles are difficult to explain, but may not want to be completely ignored. Again, as in my previous comments, this does not change the story of the manuscript, which rather focuses on the larger bend angle of approximately 35° at the AAF lesion in the absence of XPA (which is clearly not present in the non-damaged DNA) compared to 60° in the XPA-DNA complex.

We appreciate the reviewer bringing up this point. We have added acknowledgement of the discrepancy in bend angles to the discussion, in lines 324-327. If our assumption that the 35° bend corresponds to the AAF site and the 10° corresponds to the non-damaged end of the AAF₅₃₈ substrate, we believe it is possible that the 10° population was indistinguishable from the 0° population on the ND₅₃₈ substrate. Because this conclusion is largely speculative, we do not wish to over-interpret our data in the discussion. We thank the reviewer for stating that the nature of the 10° bend should not significantly change our study.

References

- 1 Yang, Z. *et al.* Specific and efficient binding of xeroderma pigmentosum complementation group A to double-strand/single-strand DNA junctions with 3'- and/or 5'-ssDNA branches. *Biochemistry* **45**, 15921-15930, doi:10.1021/bi061626q (2006).
- 2 Wood, R. D. DNA damage recognition during nucleotide excision repair in mammalian cells. *Biochimie* **81**, 39-44, doi:10.1016/s0300-9084(99)80036-4 (1999).
- 3 Sancar, A., Lindsey-Boltz, L. A., Unsal-Kacmaz, K. & Linn, S. Molecular mechanisms of mammalian DNA repair and the DNA damage checkpoints. *Annu Rev Biochem* **73**, 39-85, doi:10.1146/annurev.biochem.73.011303.073723 (2004).
- 4 Liu, Y. *et al.* Cooperative interaction of human XPA stabilizes and enhances specific binding of XPA to DNA damage. *Biochemistry* **44**, 7361-7368, doi:10.1021/bi047598y (2005).
- 5 Koch, S. C. *et al.* Structural insights into the recognition of cisplatin and AAF-dG lesion by Rad14 (XPA). *Proceedings of the National Academy of Sciences of the United States of America* **112**, 8272-8277, doi:10.1073/pnas.1508509112 (2015).
- 6 Sugawara, K. *et al.* Xeroderma pigmentosum group C protein complex is the initiator of global genome nucleotide excision repair. *Mol Cell* **2**, 223-232, doi:10.1016/s1097-2765(00)80132-x (1998).

- 7 Volker, M. *et al.* Sequential assembly of the nucleotide excision repair factors in vivo. *Mol Cell* **8**, 213-224 (2001).
- 8 Rademakers, S. *et al.* Xeroderma pigmentosum group A protein loads as a separate factor onto DNA lesions. *Mol Cell Biol* **23**, 5755-5767 (2003).
- 9 Sugitani, N., Sivley, R. M., Perry, K. E., Capra, J. A. & Chazin, W. J. XPA: A key scaffold for human nucleotide excision repair. *DNA Repair (Amst)* **44**, 123-135, doi:10.1016/j.dnarep.2016.05.018 (2016).
- 10 Sugitani, N., Voehler, M. W., Roh, M. S., Topolska-Wos, A. M. & Chazin, W. J. Analysis of DNA binding by human factor xeroderma pigmentosum complementation group A (XPA) provides insight into its interactions with nucleotide excision repair substrates. *J Biol Chem* **292**, 16847-16857, doi:10.1074/jbc.M117.800078 (2017).
- 11 Yang, Y., Sass, L. E., Du, C., Hsieh, P. & Erie, D. A. Determination of protein-DNA binding constants and specificities from statistical analyses of single molecules: MutS-DNA interactions. *Nucleic Acids Res* **33**, 4322-4334, doi:10.1093/nar/gki708 (2005).
- 12 Beckwitt, E. C., Kong, M. & Van Houten, B. Studying protein-DNA interactions using atomic force microscopy. *Semin Cell Dev Biol* **73**, 220-230, doi:10.1016/j.semcdb.2017.06.028 (2018).
- 13 Margeat, E., Le Grimellec, C. & Royer, C. A. Visualization of trp repressor and its complexes with DNA by atomic force microscopy. *Biophysical journal* **75**, 2712-2720, doi:10.1016/S0006-3495(98)77715-X (1998).
- 14 Schulz, A., Mucke, N., Langowski, J. & Rippe, K. Scanning force microscopy of Escherichia coli RNA polymerase.sigma54 holoenzyme complexes with DNA in buffer and in air. *J Mol Biol* **283**, 821-836, doi:10.1006/jmbi.1998.2131 (1998).
- 15 Chen, L., Haushalter, K. A., Lieber, C. M. & Verdine, G. L. Direct visualization of a DNA glycosylase searching for damage. *Chem Biol* **9**, 345-350 (2002).
- 16 Buechner, C. N. *et al.* Strand-specific recognition of DNA damages by XPD provides insights into nucleotide excision repair substrate versatility. *J Biol Chem* **289**, 3613-3624, doi:10.1074/jbc.M113.523001 (2014).
- 17 Josephs, E. A. *et al.* Structure and specificity of the RNA-guided endonuclease Cas9 during DNA interrogation, target binding and cleavage. *Nucleic Acids Res* **43**, 8924-8941, doi:10.1093/nar/gkv892 (2015).
- 18 Sukhanova, M. V. *et al.* Single molecule detection of PARP1 and PARP2 interaction with DNA strand breaks and their poly(ADP-ribosyl)ation using high-resolution AFM imaging. *Nucleic Acids Res* **44**, e60, doi:10.1093/nar/gkv1476 (2016).